# Study on the Biological Communities and Bioweathering of the Marble Surfaces of the Hall of Prayer for Good Harvests at the Temple of Heaven (Beijing, China)

Youping Tian *

School of Earth Sciences and Resources, China University of Geosciences (Beijing), Beijing,China

**Abstract**: This study investigates the biological communities and bioweathering of the marble surfaces of the Hall of Prayer for Good Harvests at the Temple of Heaven in Beijing. The dominant organisms are aerophytic cyanobacteria, which thrive in calcareous environments, are drought-resistant, slow-growing, and highly resilient. These cyanobacteria exhibit different community compositions depending on the orientation of the marble surface. On east-facing, warm and humid surfaces, the communities are mainly composed of small filamentous cyanobacteria such as *Scytonema* sp.2 and coccoid cyanobacteria like *Gomphosphaeria* sp. On west-facing, hot and humid surfaces, the dominant organisms are *Scytonema* sp.1 (a small filamentous cyanobacterium) and mosses. On north-facing, cold and humid surfaces, the biological communities mainly consist of coccoid cyanobacteria such as *Myxosarcina* sp. and *Gomphosphaeria* sp. On south-facing, hot and dry surfaces, the communities are primarily made up of small or large filamentous cyanobacteria, including *Scytonema* sp.1 and *Nostoc* sp. The intensity of weathering observed varies by orientation: South > West > East > North. This pattern aligns with the observed "Cloud Chi Heads" weathering features on surfaces facing different directions. The biological communities on the marble surface display a range of colors, with gray-black being the most common, followed by gray-white, black, brown, and dark brown. The gray-black communities are mainly composed of *Myxosarcina* sp. and *Gomphosphaeria* sp. These communities also exhibit various morphologies, including membranous,

---

* Corresponding author at: School of Earth Sciences and Resources, China University of Geosciences (Beijing), Beijing,China
E-mail address: typ@cugb.edu.cn (Y. Tian)

pilose, carpet-like, leathery, shell-like, and powdery layers. The species composition varies across these morphological types. The growth of aerophytic organisms on rock surfaces is controlled by macroscopic hydrodynamics and micro-topographical features. At the macro scale, in areas with low rainfall intensity, biofilms are sparse and biological weathering is weak. In areas with high rainfall, cyanobacteria-rich "ink bands" can form, leading to intense biological weathering. At the micro scale, micro-topographical features regulate local hydrological conditions and determine colonization patterns: rough, uneven surfaces and discrete water films promote spot-like bio-colonies that lead to solution pores and pits; linear decorations or joints with directional water retention drive linear biological growth, forming solution marks and grooves; smooth, dense surfaces with uniform water film coverage support planar microbial growth, ultimately resulting in overall layer separation from weathering. The mechanism of biological weathering involves the secretion of organic acids by aerophytic organisms. These acids dissolve inorganic salts in the rock, providing nutrients while gradually "eroding" the rock, damaging its surface structure, and leading to progressive weathering. Preventing or reducing the growth of aerophytic organisms is key to slowing the biological weathering of the stone relics on the Hall of Prayer for Good Harvests.

Keywords:The Hall of Prayer for Good Harvests, marble, aerophytic organisms, cyanobacteria, bioweathering

## 1. introduction

The Hall of Prayer for Good Harvest, located in the Temple of Heaven in Beijing, China, was completed in 1420. It served as the site for the emperors of the Ming and Qing dynasties to perform the "Heaven Worship" and "Prayer for Good Harvest" rituals. It is also the largest existing ancient architectural complex for heaven worship in the world. In 1961, the State Council of China designated the Temple of Heaven as a "National Key Cultural Relic Protection Unit." In 1998, it was recognized by UNESCO as a "World Cultural Heritage Site." The base of the Hall of Prayer for Good Harvests is a three-tiered circular platform made of white marble, 6 meters high and surrounded by a balustrade (Fig. 1-a). The marble used in the construction is

divided into two types: White marble and Bluish-white marble. White marble, due to its fine

texture and ease of detailed carving, is often used in decorative parts such as balustrades and

carvings. Bluish-white marble, with its higher compressive strength (Ye and Zhang, 2019) and

better corrosion resistance compared to White marble (Qu, 2018), is typically used in load-

bearing and wear-resistant areas such as the base and paving. Most of the White marble and

Bluish-white marble used in the construction come from the marble quarries in Dashiwo Town,

Fangshan District, Beijing. (Wu and Liu, 1996; Lü and Wei, 2020). Compared to other types of

rock used in the Temple of Heaven complex (such as limestone, granite, and sandstone), the

marble used in the Hall of Prayer for Good Harvests is the most susceptible to weathering due

to its lower hardness. White marble, a special variety of marble, is particularly sensitive to

weathering (Ye and Zhang, 2019). Additionally, because marble is rich in calcium, it serves as

a preferred substrate for biological growth (Miller, et al., 2006). However, the organisms are

not uniformly distributed across the entire marble surface; their distribution is selective. In

addition to requiring a calcium-rich substrate, they also need water. In areas with low rainfall

intensity (such as high and protruding locations on the marble surface), where water is scarce,

there is little to no biological growth, and the surface appears white or yellowish-white with

minimal biofilm and weak bioweathering. In contrast, in areas with high rainfall intensity (such

as water convergence points, channels, and Chi Heads), where water is abundant, there is

extensive biological growth, and the surface appears black (with patches of brown and gray-

black) with a prominent biofilm and strong bioweathering. (Fig.1-b). The gradient distribution

of the biofilm on the rock surface is significantly spatially coupled with the variations in

instantaneous runoff, reflecting an optimal water allocation mechanism in arid environments

(Tian, 2004; Macedo, et al., 2009). In addition to the rock substrate and precipitation,

environmental factors such as wind and air pollution also influence microbial colonization, a

phenomenon known as "bioreceptivity" of vulnerable structural materials (Guillitte and

Dreesen, 1995; Miller, et al., 2012). Among these organisms, cyanobacteria are particularly

significant because they can grow with just light and water, and they can survive within the

rock, playing a crucial role in the degradation of stone cultural relics (C., Gaylarde, 2020).

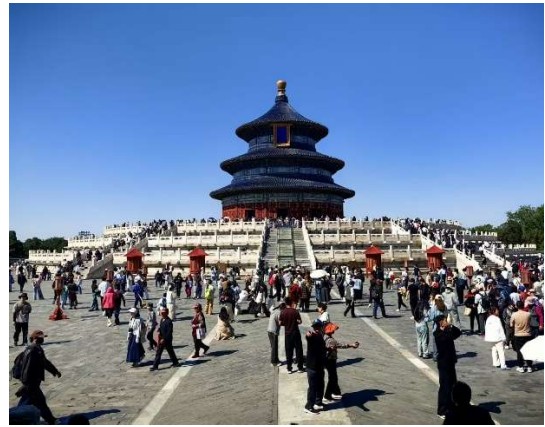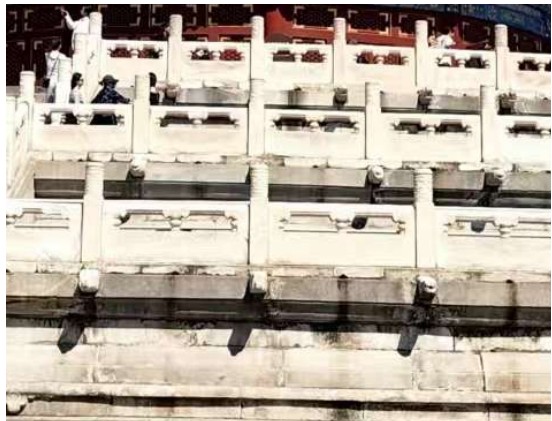

a                                                                          b

Fig. 1. The hall of Prayer for Good Harvests in the Temple of Heaven, Beijing, China, and

the distribution of black biofilm on its marble surface.

a. The Hall of Prayer for Good Harvests has a base consisting of three tiers of marble platforms; b The surface of marble platforms is

covered with black biofilm, which is distributed according to the intensity of rainfall. In areas with low rainfall intensity, the biofilm is not

noticeable. In areas with high rainfall intensity (such as channels, water convergence points, and Chi Heads), there is a significant

distribution of black biofilm.

Black microbial distributions, forming black crusts, have been observed on marble surfaces in many regions (Checcucci, et al., 2022; Monte and Sabbioni, 1986; Praderio, et al., 1993; Gorbushina, et al., 2002). This phenomenon is also referred to as marble blackening (Moropoulou, et al., 1998), bioweathering, or biodeterioration. The microbial communities on marble surfaces exhibit high diversity (Timoncini et al., 2022). The black crust microbial communities are primarily composed of coccoid and filamentous cyanobacteria from the genera *Chroococcus*, *Gloeocapsa*, and *Tolypothrix* (Lombardozzi, et al., 2012), as well as green algae, fungi (Isola, et al., 2016; Leo, et al., 2019; Marvasi, et al., 2012), and lichens (Pinna, et al., 2018). *Chroococcus* can bore into the marble surface, demonstrating remarkable environmental adaptability: not only do they form blue-green biofilms on the rock surface (epilithic growth), but they also penetrate through cracks (chasmoendolithic growth), colonize mineral interstices (cryptoendolithic growth), and even actively excavate (euendolithic growth) deeper into the marble. The tubular tunnels drilled by *Chroococcus* in calcite crystals involve both chemical dissolution and mechanical erosion, making them a dominant species in the community (Golubić, et al., 2015; Scheerer, et al., 2009). Biofilms

alter the thermal and moisture properties of the material, exert colloidal mechanical stress, and secrete acidic and redox metabolites, which intensify mineral lattice destruction and promote the formation of harmful crusts (Guiamet, et al., 2013; Warscheid and Braams, 2000). They can also accelerate rock weathering, leading to the formation of pits (Danin and Caneva, 1990) and control the micro- and macro-morphology of the rock surface (Tian, 2004). Black biofilms on marble surfaces show differential erosion based on orientation, such as differences between windward and leeward faces (Danin and Caneva, 1990). Height differences also play a role, with height having a greater impact on microbial weathering than orientation. The microenvironmental gradients on the rock surface are the core driving factors for the biological erosion of stone cultural heritage (Trovão and Portugal, 2024). In extremely arid environments, "gravel shell" microbial communities composed of lichens, cyanobacteria, and fungi drive the decomposition of rock particles and the formation of primitive soil (terrestrial protopedon) through bioweathering mechanisms such as pH changes, swelling and shrinking, enzymatic activity, and mineral migration (Jung, et al., 2020). Even in areas with fewer black biofilms, the frequency of microbial presence increases as the physical and chemical acid erosion of marble forms a powdery layer, accelerating the transformation of marble into soil and posing a serious threat to marble cultural heritage. Understanding the bioweathering patterns on marble surfaces is crucial for the conservation of marble cultural heritage. For example, targeted use of microbial methods to remove black crusts can be more effective than purely chemical or laser methods (Gioventù, et al., 2011).

Current research on the weathering of marble cultural heritage in Beijing has primarily focused on the roles of physical and inorganic chemical processes, such as acid erosion. Studies have found that the surface peeling and pollution of the marble steles at the Confucian Temple in Beijing are caused by acid rain erosion (He, 2021). Freeze-thaw cycles can lead to internal structural damage in rocks, while salt fog crystallization causes pore expansion and degradation (Li, 2023). Temperature changes affect the physical and mechanical properties of dolomitic marble in Beijing, leading to surface peeling, dissolution of dolomite crystals, and the formation of crusts due to $SO_2$ and dust pollution (Liu, 2020; Zhang, et al., 2016; Wang, et al., 2022). The mechanism of granular peeling on marble surfaces is attributed to the low amount of cementing material between particles, resulting in weak cementation. Surface particles are disrupted by mining unloading, processing damage, stress concentration, and temperature variations, leading to peeling (Wang, 2010).

Temperature fluctuations, acid rain dissolution, water erosion, and salt micro-crack filling are the main causes of weathering in Fangshan marble in Beijing (Zhang, et al., 2015). Research on the weathering of white marble components in the Hall of Supreme Harmony in the Forbidden City indicates that thermal stress from solar radiation and rain erosion are the primary factors (Wu, et al., 2023). It has been found that under the combined action of acid and salt, dolomite crystals degrade through dissolution, interstitial erosion, and spalling. Salt crystallization accelerates the latter two types of damage, while acid erosion promotes salt penetration, significantly increasing the rate of degradation (Zheng, et al., 2025). Regarding bioweathering of marble in Beijing, only a few studies have mentioned it (Beijing Institute of Ancient Architecture, 2018). The main types of damage to marble in Beijing include fissures, peeling, disintegration, crust formation, solution pits, erosion, component loss, discoloration, biological parasitism, and improper human restoration (Yang, 2016). Two types of peeling in white marble in the Beijing area have been identified: one driven by the synergistic effects of thermal weathering, lichen, and rainfall, and the other by acid rain and capillary water absorption (Zhang, 2022). To understand the patterns of bioweathering in marble, it is essential to know the composition, structure, and metabolic potential of the resident microbial communities and their interactions with the stone (Pinna, 2017; Marvasi, et al., 2019). This study focuses on analyzing the community composition, structure, and relative biomass of biofilms on the marble surface of the Hall of Prayer for Good Harvests in Beijing. By identifying the development process and patterns of the biofilm communities, we aim to reveal the mechanisms of biocorrosion and provide a scientific basis for developing more targeted conservation strategies for marble cultural heritage.

1. Overview of the Study Area

Beijing is located in a warm temperate monsoon semi-humid climate zone, characterized by a cool mountain climate. The region has an average annual temperature of 10.8°C, with a frost-free period of approximately 150 days. In winter, Beijing is influenced by cold air masses from the northwest, resulting in a cold and dry climate. The prevailing wind direction during this season is from the northwest, with an average annual wind speed of 1.9 meters per second. In summer, the influence of the subtropical high-pressure system makes the climate hot, and rainfall is relatively

concentrated, especially from July to September, when about 85% of the annual precipitation occurs,
often in the form of heavy rain. Autumn in Beijing is generally pleasant, while spring is relatively
short. The frost-free period ranges from 190 to 200 days. Under extreme weather conditions, the
maximum summer temperature can reach 42°C, and the minimum winter temperature can drop to -
25°C. According to data from the National Meteorological Information Center of Beijing from 2009
to 2024, the annual precipitation in Beijing shows significant fluctuations (Fig. 2). There is no clear
trend in the annual average relative humidity and annual average precipitation, but there is an
increasing trend in the annual average sunshine hours and annual average extreme maximum
temperature, and a decreasing trend in the annual average extreme minimum temperature (Fig. 3).
During the period from 2009 to 2024, the multi-year average annual total rainfall was 610 mm.

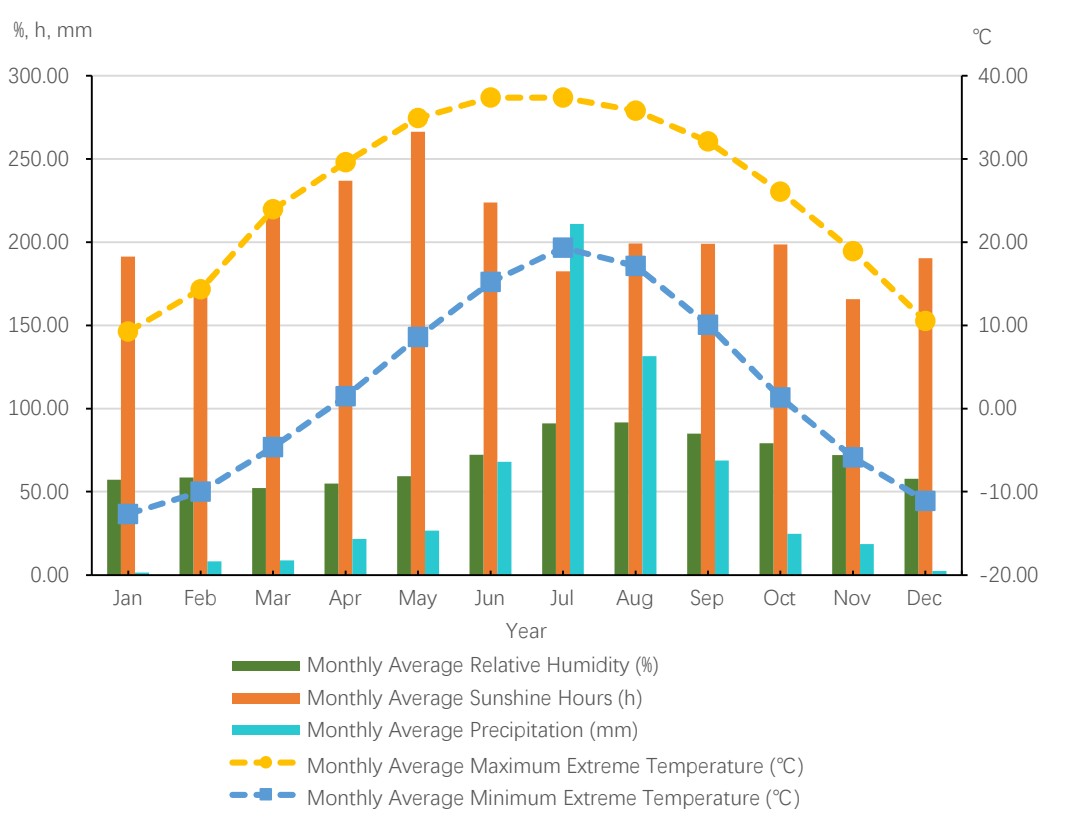

Fig. 2 Monthly average relative humidity, monthly average sunshine hours, monthly average precipitation,
monthly average extreme maximum temperature, and monthly average extreme minimum temperature
in the Beijing region from 2009 to 2024
(based on data from the National Meteorological Science Data Center Website).

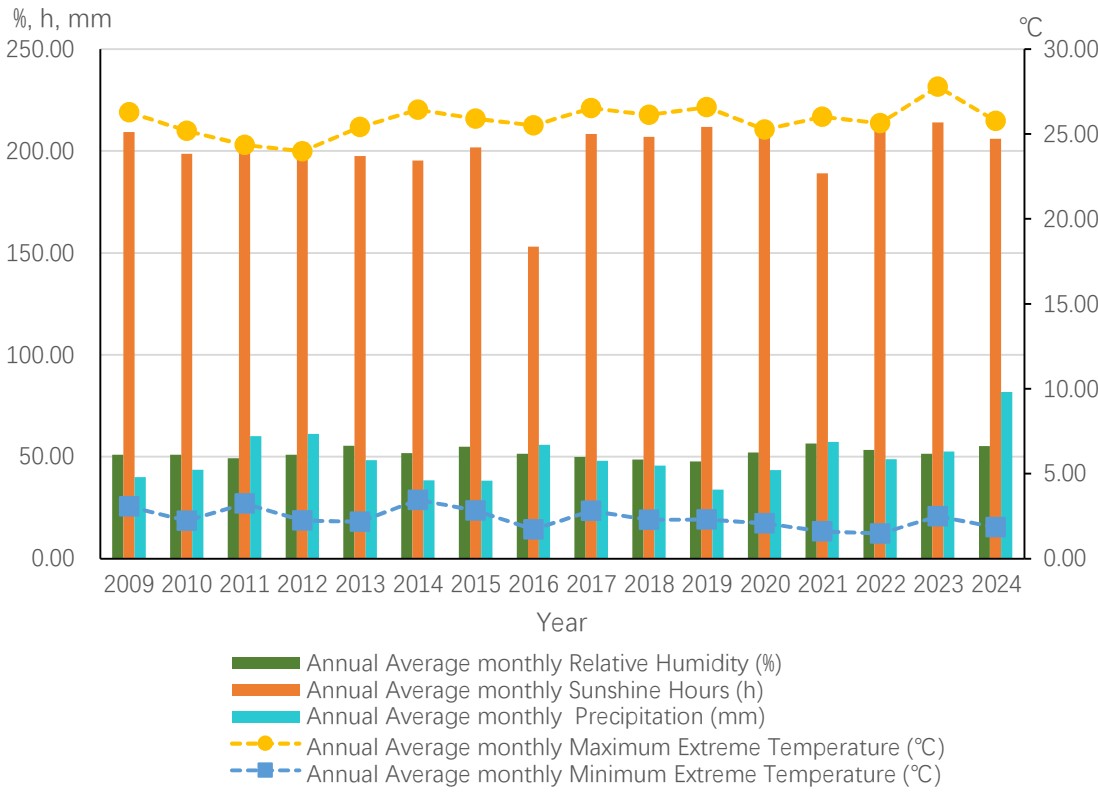

Fig. 3 Annual average relative humidity, annual average sunshine hours, annual average precipitation, annual

average extreme maximum temperature, and annual average extreme minimum temperature

in the Beijing region from 2009 to 2024

(based on data from the National Meteorological Science Data Center Website).

The base of the Hall of Prayer for Good Harvests at the Temple of Heaven is divided into three tiers (Fig. 1-a). Each tier of its white marble base is adorned with 100 intricately carved Chi Heads (Fig. 1-b, Fig. 16). The Chi Head (Chi Shou) is a unique functional and decorative architectural element in traditional Chinese architecture, inspired by the mythical hornless dragon-like creature "chi" Resembling a dragon's head without horns, it is commonly found on the roofs, beams, columns, and stone railings of palaces and temples. Its design integrates both aesthetics and practicality: rainwater is channeled through hidden drainage holes in the Chi Heads, preventing water erosion of the base while creating a distinctive visual effect. The three tiers of the altar collectively have a total of 300 Chi Heads, with the decorative themes progressively layered—dragon heads (Dragon Chi

Heads) on the upper tier symbolize imperial authority, phoenix heads (Phoenix Chi Heads) on the middle tier represent auspicious harmony, and cloud patterns (Cloud Chi Heads) on the lower tier reflect the connection between heaven and earth. During the rainy season, water cascades from the mouths of the Chi Heads on all three tiers, creating a spectacular sight of "dragons spouting torrents, phoenixes holding pearls, and clouds rolling like waves." Over time, the weathering of the Chi Heads has varied significantly depending on their orientation (Fig. 16), vividly illustrating the dynamic interaction between ancient architectural elements and the natural environment.

The main production area for Beijing marble is Dashiwo Town, located in the southwestern part of Fangshan District, Beijing. In the distribution of marble layers in Fangshan, Bluish-white marble is the first to be quarried due to its shallow burial depth. On the other hand, White marble is found in the deepest layer, with a burial depth that is usually the deepest among the stone layers, ranging from 90 cm to 150 cm in thickness. In the construction industry, both White marble and Bluish-white marble are widely used as marble materials.

2 Research Methods

2.1 Field Work

Different forms of biofilm communities on the marble surface were collected (biofilms are loose and easily detachable, so a small amount was gently picked by hand without damaging the cultural relics), placed in specimen boxes, numbered, and photographed. The appearance, color, and morphology of the biofilms were described, and the date and location were recorded. The micro-morphologies formed by the dissolution of the biofilm communities were observed and photographed. A total of 40 biofilm community specimens were collected, and 22 photographs of the field biofilm communities were taken. On clear, sunny days, the surface temperature of the rock in the sampling area was measured using an infrared thermometer (DL333380, Deli, China). At the same time, the degree of weathering of the Chi Heads was marked on the overhead view of the Hall of Prayer for Good Harvests. Chi Heads with complete surface ornamentation were marked in green, those with incomplete ornamentation were marked in yellow, and those with completely weathered and disappeared ornamentation were marked in red. Environmental humidity in different directions was measured using a hygrometer (THM-H1, Delixi, China).

2.2 Laboratory Work
2.2.1 Microscopic Observation
The size, morphology, and color of the biofilm communities were observed using a
stereomicroscope (Szx7, Olympus, Japan). Then, temporary slides were prepared from different
colored biofilm communities and observed under a biological microscope (Bx51, Olympus, Japan).
The species of the biofilm communities were identified (Desikachary, 1959; Geitler, 1932; Komarek,
1998, 2005, 2013; Zhu, 1991; Fott, 1980; Hu and Wei, 2006), and photographs were taken. For each
biofilm community ecological specimen, a microslide was prepared, resulting in a total of 40
microslides, and 142 microscopic photographs were taken.
2.2.2 Biomass Statistics
The volume percentage of the species in the biofilm communities was recorded. The volume
percentages of the species were statistically analyzed to calculate the relative volume (Vx, relative
biomass) and the relative volume percentage (Yx, relative biomass percentage). The statistical and
calculation methods are as follows:
(1) Relative Volume (Vx, Relative Biomass)
To obtain the relative volume, the following steps need to be taken:
1) Determine the volume percentage (v(x)%)
By estimating the percentage of the volume that each species occupies relative to the total
volume of all species in each microslide, the volume percentage (v(x)%) of that species in the
microslide is obtained. The estimation can be based on the area occupied by each species in the
microslide, as within the same microslide, the thickness between the cover slip and the slide is nearly
uniform across different areas. Therefore, under the same thickness, the larger the area occupied by
a species, the greater its volume (Fig. 4).

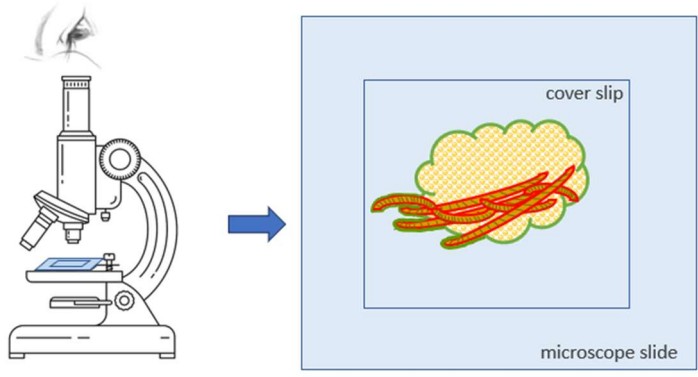

$$V(Specie1)\% = \frac{specie1\ area \times high}{(specie1\ area +\ specie2\ area) \times high} \times 100\,\%$$

V(Specie1)%: volume percentage of specie1

area within the red line: specie1 area
area within the green line: specie1 area + specie2 area

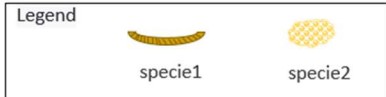


Fig. 4 Visual Method for the volume percentage (v(x)%) of a Specie

in a Biofilm Community Microslide

If there are two species in the microslide: Species 1 and Species 2, the volume percentage of Species 1 can be estimated by dividing the
area occupied by Species 1 by the total area occupied by both Species 1 and Species 2, and then multiplying by 100. This gives the volume
percentage (v(x)%) of Species 1 in the microslide.

2) Sum the Volume Percentages
Add up the volume percentages of the same species across all microslides in the study area to
obtain the relative volume (Vx) of a species in the study area. The relative volume of a species
roughly reflects its relative biomass in the biofilm community of the study area. It does not represent
the actual volume but is an estimated relative value that is meaningful for comparison.

$$Vx = v(x)_{i_1} + v(x)_{i_2} + \cdots + v(x)_{i_x}$$

$i_x$ is the microslide number; x is a specific species(x=a,b,c,…); $v(x)_{ix}$% is the volume percentage of
species x.
(2) Relative Volume Percentage (Yx, Relative Biomass Percentage)
This is the percentage of the relative volume (Vx) of a species in the biofilm community of the

study area relative to the sum of the relative volumes of all species in the biofilm community. It is

also referred to as the relative biomass percentage.

$$Y_x = \frac{V_x}{n \times 100} \times 100\%$$

n is the total number of microslides.

The relative volume percentage, also known as the relative biomass percentage, does not represent the actual biomass. This is because it is currently very difficult to accurately measure the biomass of biofilm communities on marble rock surfaces. By estimating through microscopic observation, one can get a rough understanding of the growth status of the biofilm community. It is a relative value and is meaningful only for comparative purposes.

3 Results

3.1 Distribution of communities in the Study Area

The composition of the biofilm communities on the marble surface in the study area includes a total of 30 genera and species (Fig. 5). The most abundant species is *Myxosarcina* sp., followed by *Gomphosphaeria* sp., *Asterocapsa* sp., *Gloeocapsa* sp.1 (Fig. 6), and *Scytonema* sp.1, among others.

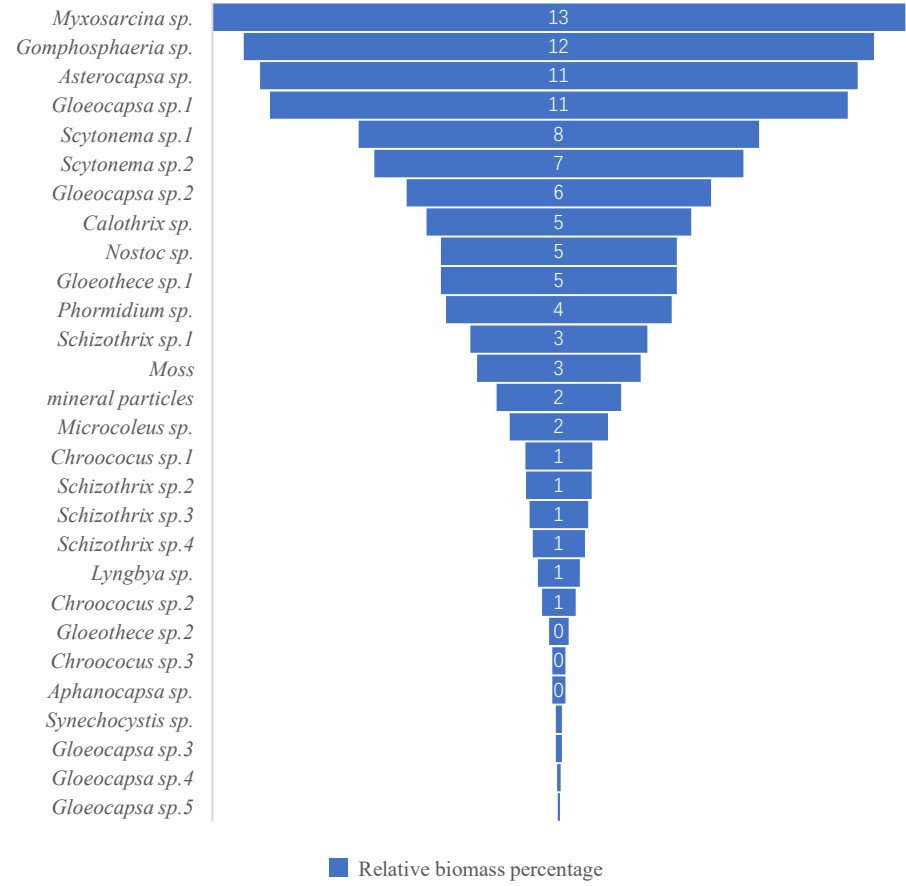

Relative biomass percentage


Fig. 5. Relative biomass percentage of marble surface of the Hall of Prayer for Good Harvests


in the Temple of Heaven, Beijing. China.



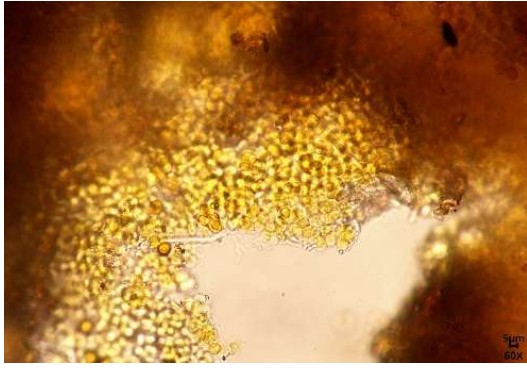 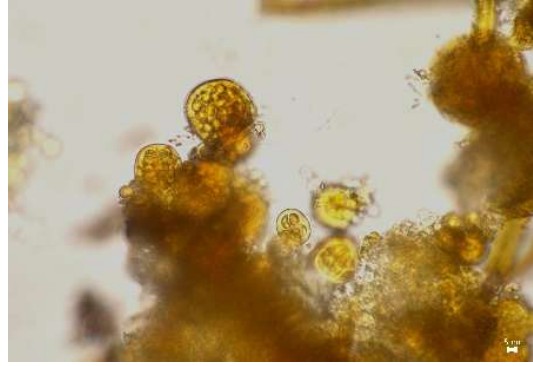

**a.** *Myxosarcina* sp.        **b.** *Gomphosphaeria* sp.

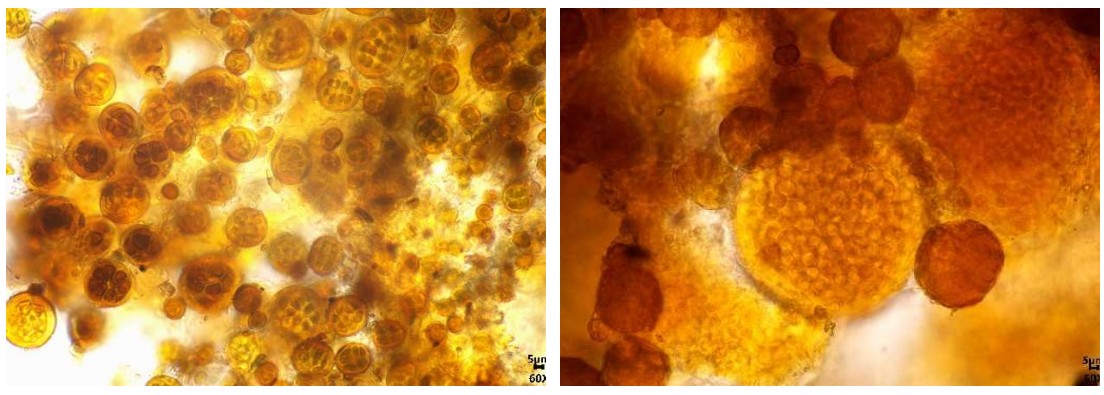

**c.** *Asterocapsa* sp.             **d.** *Gloeocapsa* sp.1

**Fig. 6.** Dominant organisms on the marble surface of the Hall of Prayer for Good Harvests

in the Temple of Heaven, Beijing., China.


3.2 Characteristics of the distribution of biological communities on marble surfaces with

different orientations in the study area

The Hall of Prayer for Good Harvests at the Temple of Heaven is a circular building (Fig. 16).

The marble surfaces facing different directions receive varying amounts of sunlight. The south-

facing surface receives the most sunlight, followed by the east and west-facing surfaces, which

receive sunlight for half a day. The north-facing surface is shaded and receives no direct sunlight.

This variation in sunlight exposure leads to differences in the biological populations on the rock

surfaces. The details are discussed below:

3.2.1 Characteristics of Biological communities on East-Facing Rock Surfaces

The biological communities on east-facing rock surfaces are primarily characterized by gray-

white, gray-brown, brown, gray-black, black-brown, white, and black-brown leathery appearances.

The main species include *Scytonema* sp.2, *Chlorococcum* sp., *Gloeocapsa* sp.2, *Gloeothece* sp.1,

*Myxosarcina* sp., *Phormidium* sp., *Calothrix* sp., *Gloeothece* sp.2, *Lyngbya* sp., *Gloeocapsa* sp.5,

and *Chroococcus* sp.1 (Fig.7) . Among these, the dominant species are *Scytonema* sp.2,

*Chlorococcum* sp., accounting for 25% and 23% of the relative biomass percentage(Fig. 8).



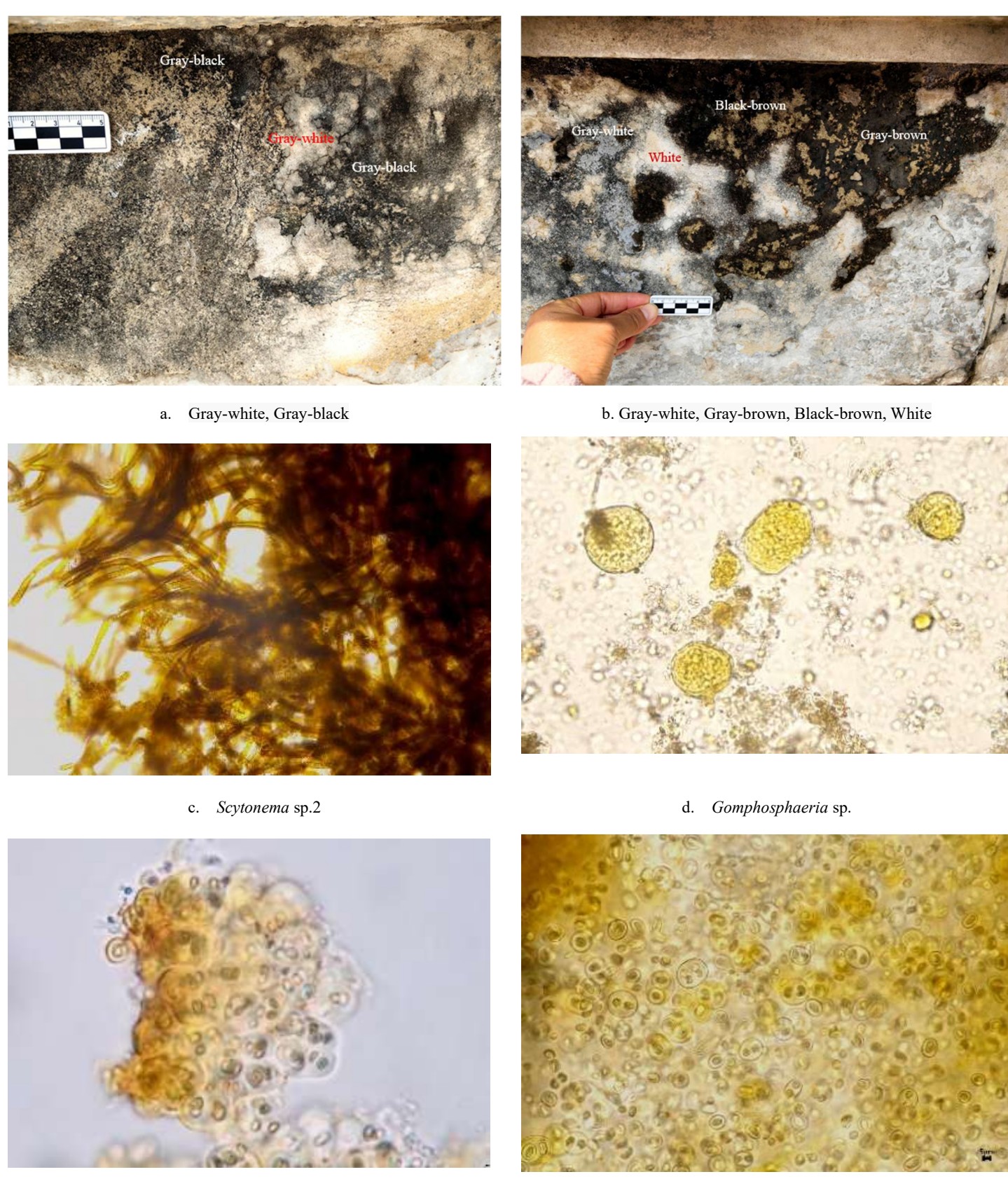

a.  Gray-white, Gray-black

b. Gray-white, Gray-brown, Black-brown, White

c.  *Scytonema* sp.2

d.  *Gomphosphaeria* sp.

e.  *Gloeocapsa* sp.2

f.  *Gloeothece sp.1*

**Fig. 7.** Micrographs of biological communities and some species on the east-facing marble surface of the Hall of

Prayer for Good Harvests in the Temple of Heaven, Beijing, China.

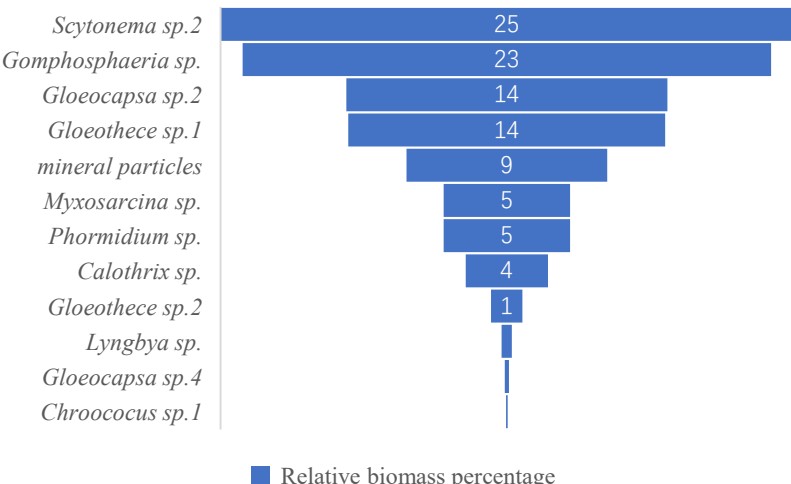

| Species | Relative biomass percentage |
|---|---|
| Scytonema sp.2 | 25 |
| Gomphosphaeria sp. | 23 |
| Gloeocapsa sp.2 | 14 |
| Gloeothece sp.1 | 14 |
| mineral particles | 9 |
| Myxosarcina sp. | 5 |
| Phormidium sp. | 5 |
| Calothrix sp. | 4 |
| Gloeothece sp.2 | 1 |
| Lyngbya sp. | |
| Gloeocapsa sp.4 | |
| Chroococus sp.1 | |

■ Relative biomass percentage


**Fig.8.** Biological population relative biomass percentage on the east-facing marble surface of the altar of
Prayer for Good Harvest in the Temple of Heaven, Beijing, China.

3.2.2 Characteristics of Biological communities on West-facing Rock Surfaces
The biological communities on west-facing rock surfaces are primarily characterized by black
hairy, black membranous, yellow-green leathery, gray-black leathery, yellow-green, brown, and
gray-green appearances. The main species include *Scytonema* sp.1, mosses, *Schizothrix* sp.1,
*Myxosarcina* sp., *Asterocapsa* sp., *Gloeocapsa* sp.1, *Gomphosphaeria* sp., and *Gloeocapsa* sp.2 et
al(Fig. 9). Among these, the dominant species are *Scytonema* sp.1 and mosses et al, accounting
for 28% and 20% of the relative biomass percentage respectively(Fig. 10).

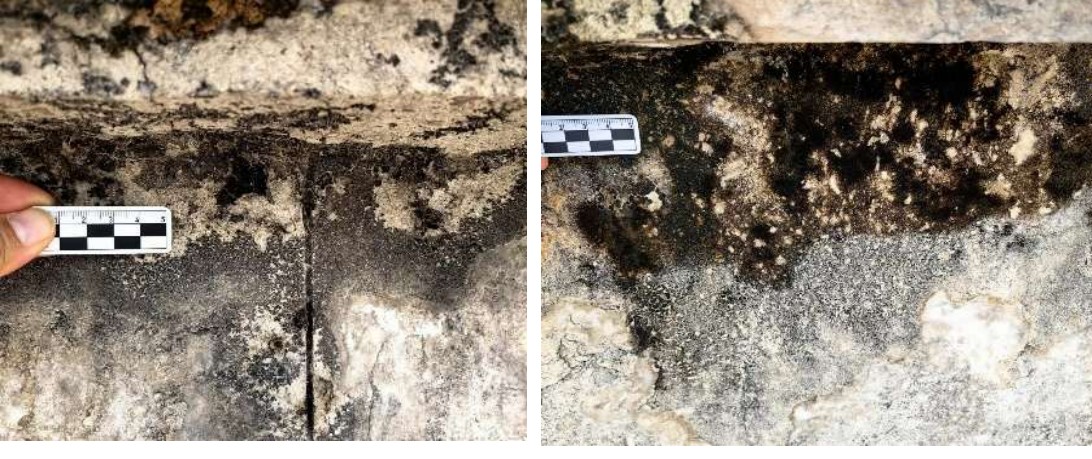

a. Gray-black leathery          b. Black hairy and membranous

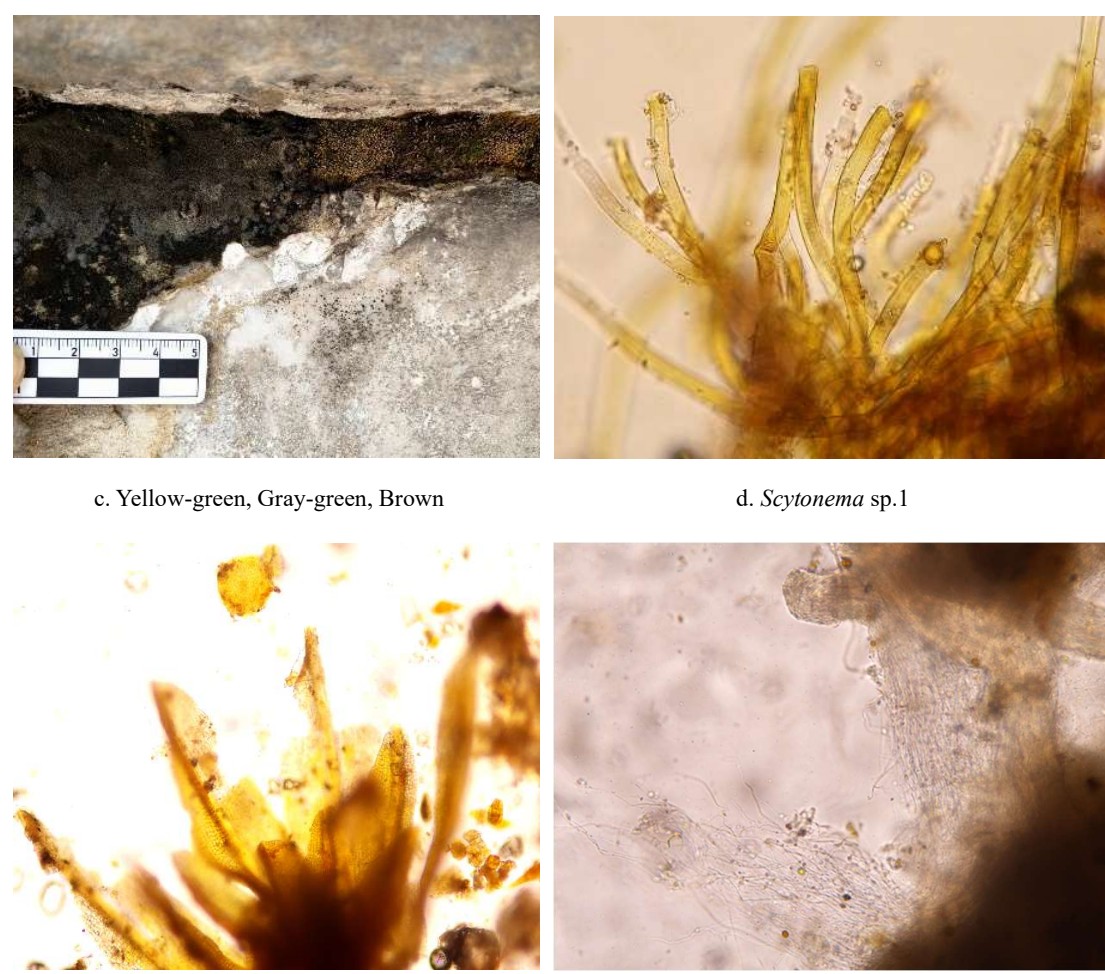

| c. Yellow-green, Gray-green, Brown | d. *Scytonema* sp.1 |
|---|---|

| e. moss | f. *Schizothrix* sp.1 |
|---|---|

**Fig. 9.** Micrographs of biomes and some species on the west-facing marble surface of the altar of
Prayer for Good Harvest in the Temple of Heaven, Beijing, China.

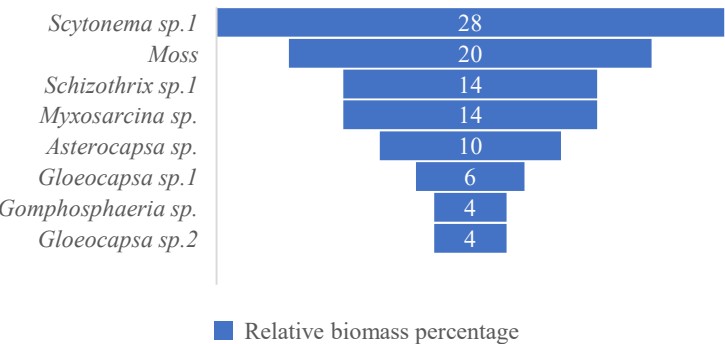



**Fig. 10.** Biological population relative biomass percentage on the west facing marble surface of the altar of
Prayer for Good Harvest in the Temple of Heaven, Beijing,China.
3.2.3 Characteristics of communities Distribution on North-facing Surfaces

The biological communities on north-facing rock surfaces are primarily characterized by gray-

brown membranous, brown, gray-black, yellow-green, black-brown, gray-white, brown crusty,
brown carpet-like, brown-black leathery, and brown-black membranous appearances. The main
species include *Myxosarcina* sp., *Gomphosphaeria* sp., *Gloeocapsa* sp.1, *Schizothrix* sp.1,
*Asterocapsa* sp., *Scytonema* sp.1, *Calothrix* sp., mosses, *Gloeocapsa* sp.2, *Microcoleus* sp.,
*Chroococcus* sp., *Gloeothece* sp.1, *Lyngbya* sp., *Gloeocapsa* sp., *Scytonema* sp.2, and *Synechocystis*
sp. et al(Fig. 11). Among these, the dominant species are *Myxosarcina* sp. and *Gomphosphaeria*
sp., accounting for 17% and 15% of the relative biomass percentage respectively(Fig. 12).


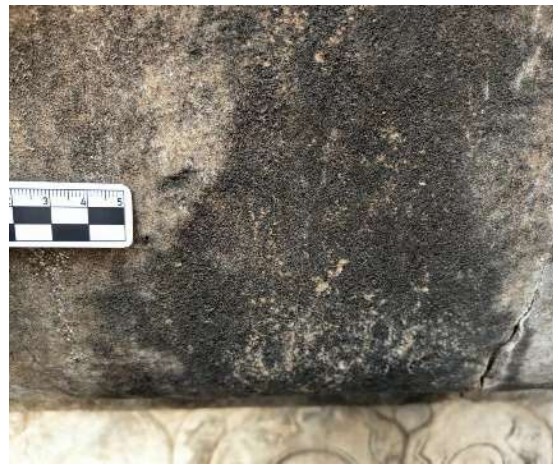

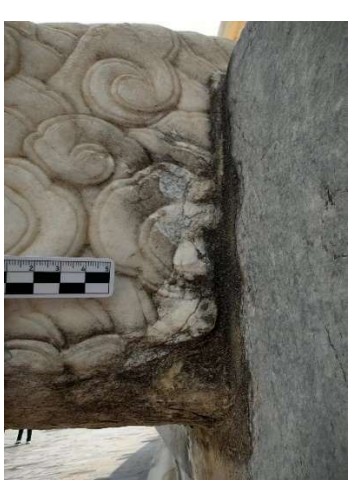

a. Gray-brown membranous                         b. Gray-black

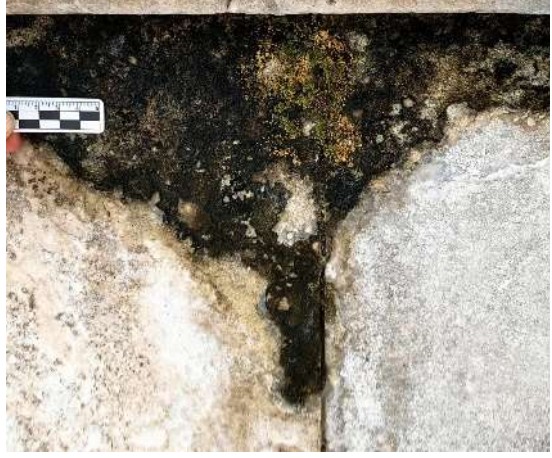

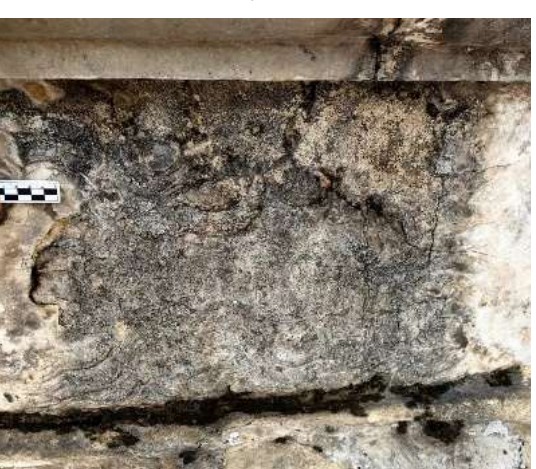

c. Yellow-green                                  d. Gray-white

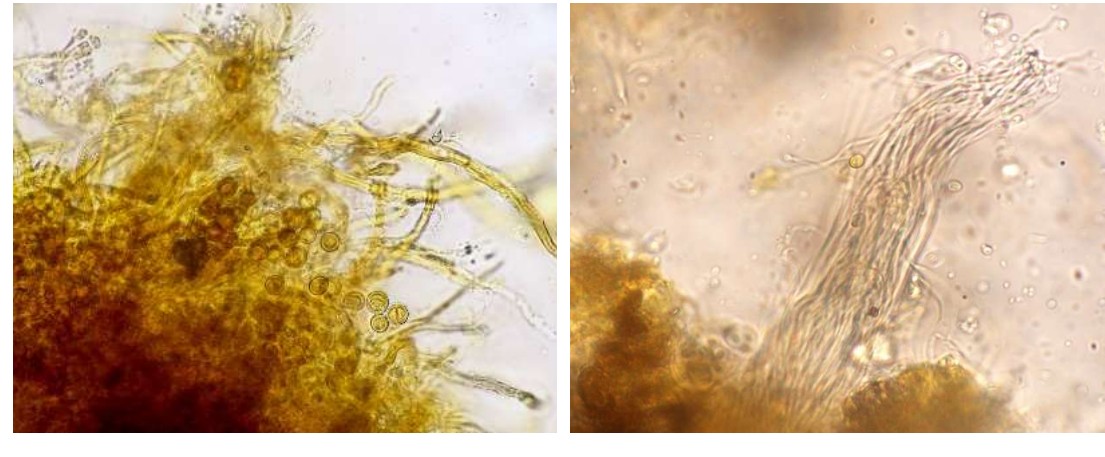

e. *Calothrix* sp.                                    f. *Microcoleus* sp.

**Fig. 11.** Micrograph of biological communities and some species on the north facing marble surface of the altar of
Prayer for Good Harvest in the Temple of Heaven, Beijing, China.

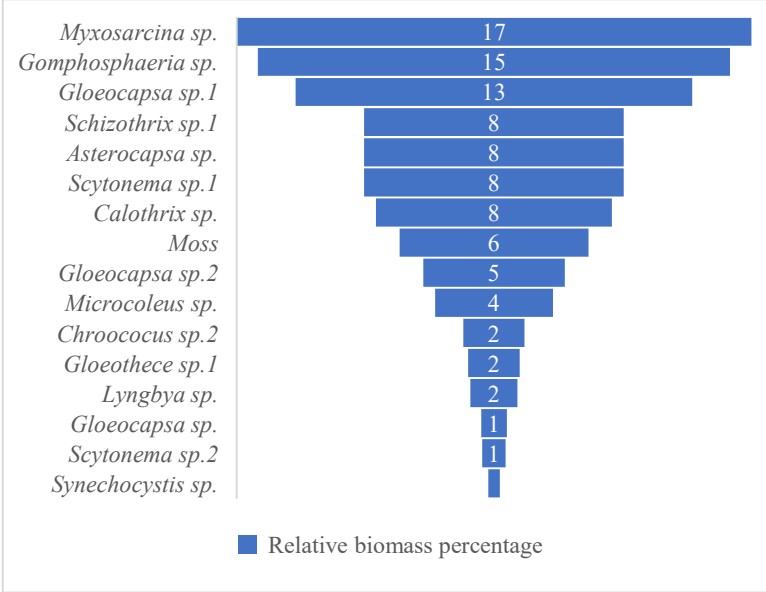

**Fig. 12.** Relative biomass percentage of biological population on the north facing marble surface of the altar of
Prayer for Good Harvest in the Temple of Heaven, Beijing,China.

3.2.4 Characteristics of communities Distribution on South-facing Surfaces

The biological communities on south-facing rock surfaces are primarily characterized by gray-
green leathery, gray-white, gray-black membranous, black leathery, gray-black, brown-yellow, and
green powdery layer appearances. The main species include *Scytonema* sp.1, *Nostoc* sp.,

*Asterocapsa* sp., *Myxosarcina* sp., *Phormidium* sp., *Gloeocapsa* sp.1, *Chroococcus* sp.1, *Schizothrix*
sp.4, *Microcoleus* sp., *Aphanocapsa* sp., *Chroococcus* sp.3, *Lyngbya* sp., *Gloeocapsa* sp.3, and
*Gloeocapsa* sp.4 et al(Fig. 14). Among these, the dominant species are *Scytonema* sp.1 and *Nostoc*
sp., accounting for 25% and 20% of the relative biomass percentage respectively(Fig. 15).

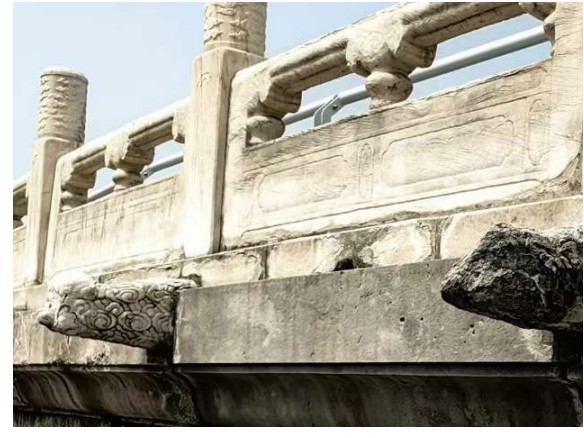

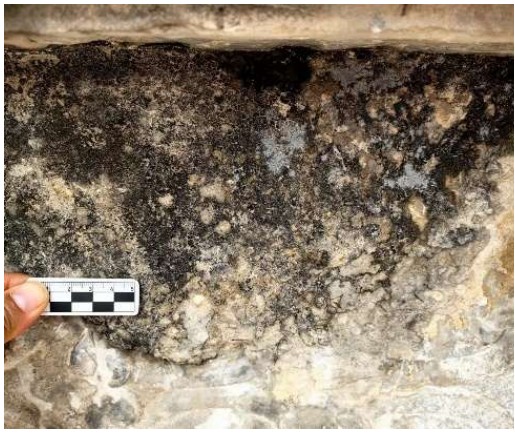

a. Some carved decorations have been completely          b. Gray-white, Gray-black

destroyed

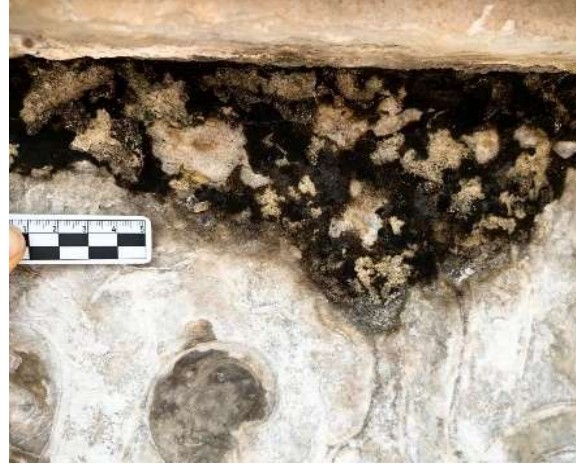

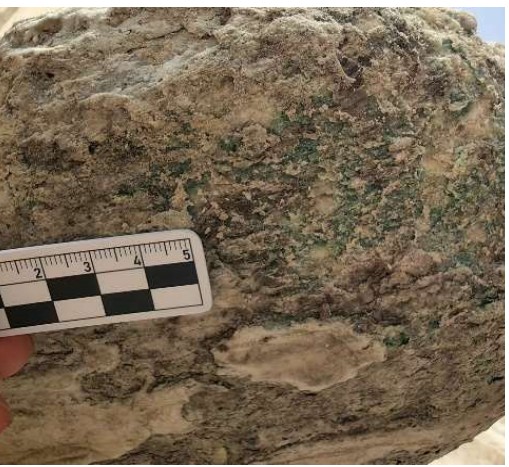

c. Black leathery          d. Green powdery layer

**Fig. 13.** Field photo of biomes on the south facing marble surface of the altar of
Prayer for Good Harvest in the Temple of Heaven, Beijing, China.


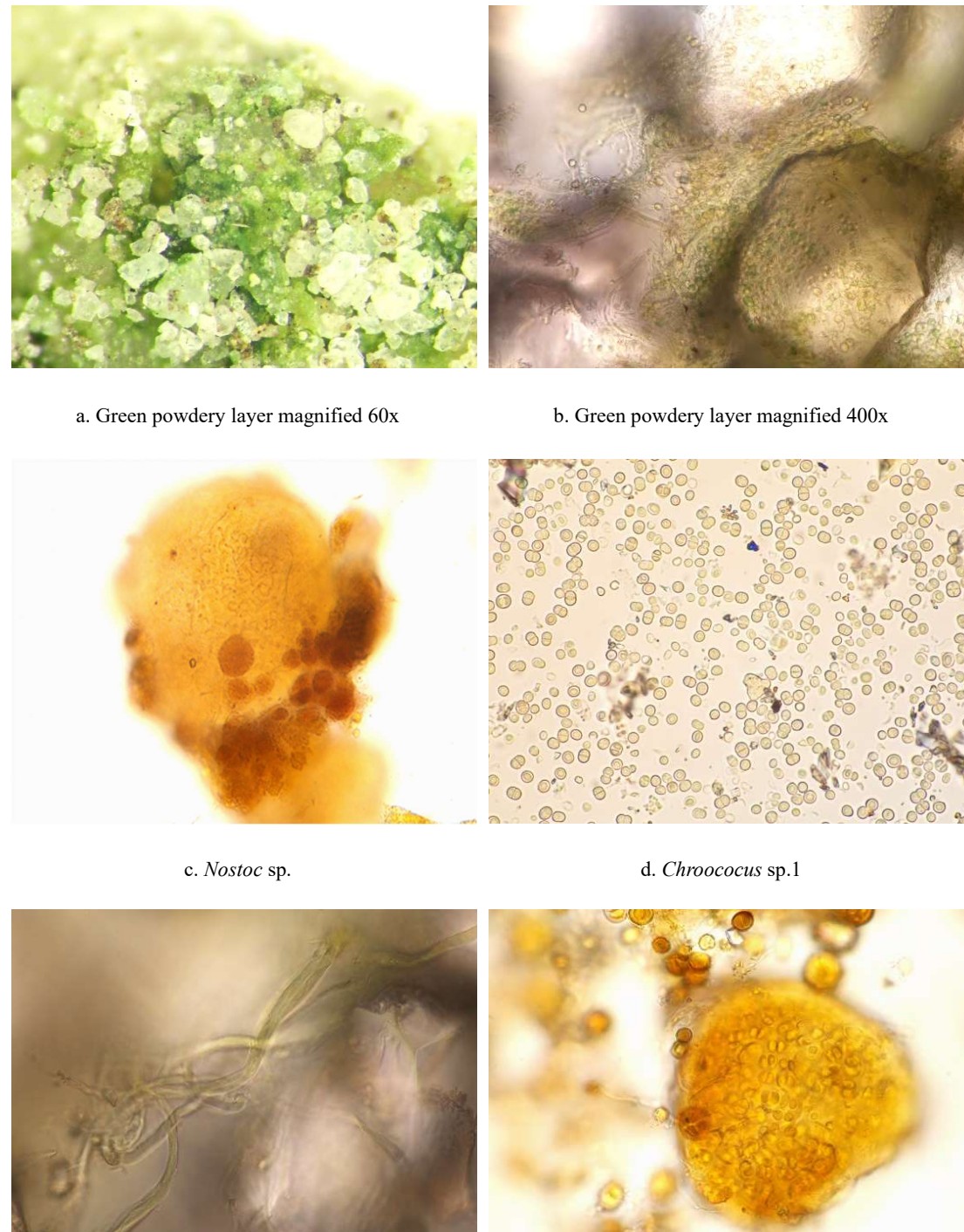

a. Green powdery layer magnified 60x      b. Green powdery layer magnified 400x

c. *Nostoc* sp.      d. *Chroococus* sp.1

e. *Schizothrix* sp.3      f. *Chroococcus* sp.3

**Fig. 14.** Micrograph of biomes and some species on the south facing marble surface of the altar of
Prayer for Good Harvest in the Temple of Heaven, Beijing, China.


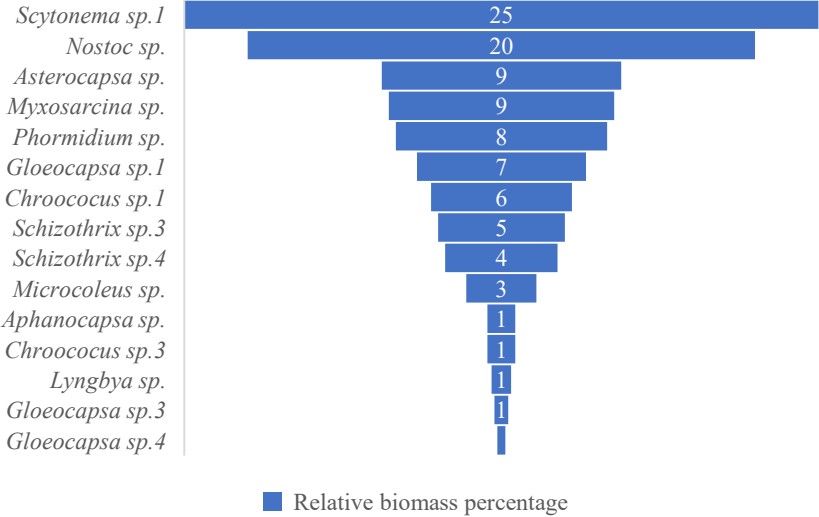

**Fig. 15.** Biological population relative biomass percentage on the south facing marble surface of the altar of

Prayer for Good Harvest in the Temple of Heaven, Beijing, China.

3.2.5 Comparison of communities on Different Orientations

The main aerophytic organisms on the rock surfaces include spherical cyanobacteria, small filamentous cyanobacteria, and large filamentous cyanobacteria. Their distribution is primarily influenced by the looseness of the substrate, sunlight, and moisture. From spherical cyanobacteria to small filamentous cyanobacteria and then to large filamentous cyanobacteria, the requirement for substrate looseness increases, the need for moisture decreases, and the requirement for sunlight duration increases. Mosses, on the other hand, prefer shady and moist environments.

Although both the east and west-facing surfaces of the Hall of Prayer for Good Harvests in the Temple of Heaven receive sunlight for half a day (Table 1), the east-facing surface receives sunlight in the morning when the rock surface temperature is lower. Even with sufficient sunlight, the growth of organisms on the east-facing surface is not as robust as on the west-facing surface. The west-facing surface receives sunlight in the afternoon when the rock surface temperature is higher, providing both water and heat conditions that are more favorable for biological growth. This results in the presence of *Scytonema* sp.1, a cyanobacterium that prefers looser substrates, and more mosses, leading to more severe weathering on the west-facing surface. The north-facing surface, being in the shade, has slower evaporation rates and is mainly colonized by spherical cyanobacteria, resulting

in relatively weaker weathering. The south-facing surface receives more sunlight and weathers faster, with the carved decorations on the rock surface completely destroyed (Fig. 13a). The matrix is highly loose, and even large filamentous cyanobacteria like *Nostoc*, which typically prefer to live in soil rather than on rock surfaces, are present. This indicates that the south-facing marble has weathered severely, forming a loose, soil-like thick weathering layer. Additionally, *Scytonema* sp.1, a species that thrives in sunny and dry environments and plays a significant role in bioweathering, is also present. Mosses are not found on the south-facing side because they prefer shady and moist environments. The orientation of the building, through differences in sunlight duration and evaporation rates, creates a unique gradient of microhabitats, which in turn drives the differential distribution of microbial communities and is accompanied by varying degrees of weathering depending on the direction.

To further understand the environmental differences and weathering conditions of the rock surfaces at the Hall of Prayer for Good Harvests in the Temple of Heaven, temperature measurements were taken on a sunny afternoon in April (Fig. 16). The rock surface temperatures were found to be highest in the southwest and lowest in the northwest. The Hall of Prayer for Good Harvests was divided into four natural sectors, each centered on a cardinal direction and covering 45° to either side: North (N): 315°-45°, centered on true north, covering from northwest to northeast; East (E): 45°-135°, centered on true east, covering from northeast to southeast; South (S): 135°-225°, centered on true south, covering from southeast to southwest; West (W): 225°-315°, centered on true west, covering from southwest to northwest. The weathering degree of 100 Cloud Chi Heads on the third layer was statistically analyzed in each sector. The results showed that 40% of the south-facing Cloud Chi Heads decorations were completely weathered, indicating the most severe weathering. The weathering degrees for the west, east, and north sectors decreased in that order. This pattern is consistent with the distribution of biological organisms on the rock surfaces, as shown in Table 1. The analysis of the weathering degree of 100 Cloud Chi Heads on the third layer showed that 40% of the south-facing Cloud Chi Heads decorations were completely weathered, indicating the most severe weathering. The weathering degrees for the west, east, and north directions decreased in that order. This pattern is consistent with the differences in weathering in different directions revealed by the distribution of biological organisms on the rock surfaces (Table 1).

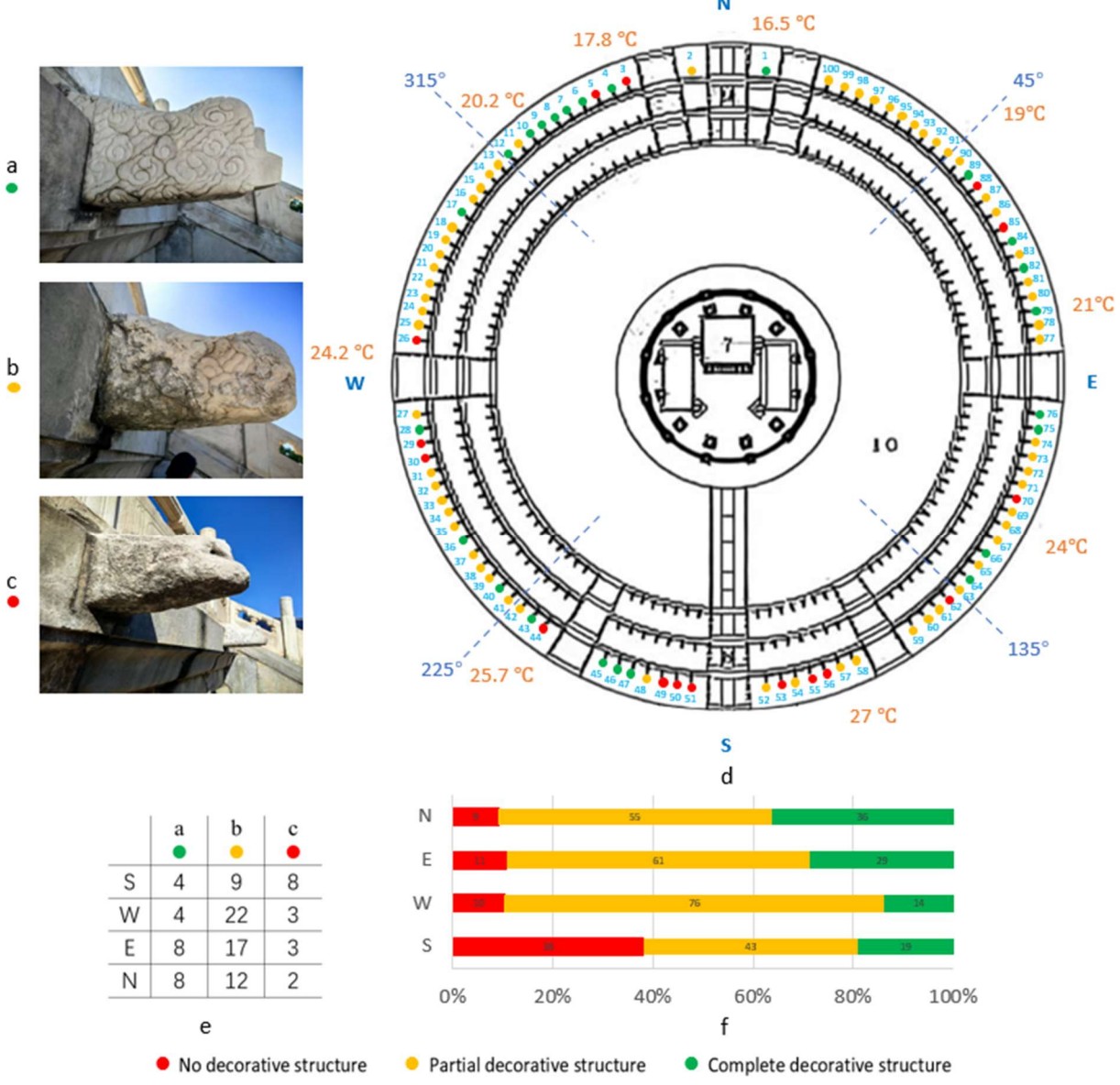


Fig. 16 Rock surface temperatures and weathering conditions of the Cloud Chi Heads on the third layer of the
Hall of Prayer for Good Harvests at the Temple of Heaven in Beijing, China, on a sunny afternoon in April.
a: Cloud Chi Heads with complete decorative structures.
b: Cloud Chi Heads with partially weathered decorative structures.
c: Cloud Chi Heads with completely weathered decorative structures.
d: A simplified top view of the Hall of Prayer for Good Harvests shows that its base is divided into three tiers, with 100 Chi Heads
arranged along the edge of each tier. The Chi Heads exhibit different degrees of weathering. This article has documented the weathering of
the outermost layer of Cloud Chi Heads: Red indicates that the decorative structure of Cloud Chi Heads is completely weathered; Yellow
indicates that the decorative structure of Cloud Chi Heads is partially weathered; Green indicates that the decorative structure of Cloud Chi
Heads is still intact.
e: Statistical count of the number of Cloud Chi Heads with three different weathering degrees in four directions.
f: Calculation of the proportion of the three different weathering degrees of Cloud Chi Heads in different directions, revealing that the
weathering intensity of the Chi Heads is highest in the south, followed by the west, east, and north.
**Table 1**
Environmental characteristics and dominant species of marble surface of the Hall of Prayer for Good
Harvests at the Temple of Heaven in Beijing, China.

| Marble Surface Orientation | Sunlight | Moisture | Environmental Characteristics | Dominant Species | Weathering degree |
|---|---|---|---|---|---|
| North-facing | None | Slow evaporation | Cold and humid | Spherical cyanobacteria | |
| East-facing | Half day | Rapid evaporation in the morning | Warm and humid | Small filamentous cyanobacteria, Spherical cyanobacteria | Weak ↑ |
| West-facing | Half day | Rapid evaporation in the afternoon | Hot and humid | Small filamentous cyanobacteria, Mosses | |
| South-facing | Full day | Rapid evaporation during the day | Hot and dry | Small filamentous cyanobacteria, Large filamentous cyanobacteria | Strong ↓ |


**3.3** Relative Biomass of Different Colored Biological Communities on Rock Surfaces in the Study
Area
The colors displayed by organisms on rock surfaces differ from those observed under a
microscope. In this paper, the former is referred to as the "visual color," while the latter is called the
"microscopic color." The visual color is the community color presented when different populations
aggregate together, whereas the microscopic color is the color of different species observed under
magnification through a microscope. Often, communities of cyanobacteria with different
microscopic colors appear mostly black or gray-black of visual color.

The visual colors of biological communities on rock surfaces in the study area can be categorized into gray-black, gray-white, black, brown, black-brown, gray-brown, yellow-green, gray-green, white, green, and brown-yellow. Their relative biomass is shown(Fig. 17). The most common color is gray-black, followed by gray-white, black, brown, and black-brown. These are also typical colors exhibited by aerophytic cyanobacteria in the field, sometimes referred to as "ink bands." For example, the Nine Horses Fresco Hill (Jiuma Huashan) in the Guilin landscape of China is formed due to aerophytic cyanobacteria growing on the rocks, creating black ink-like bands.

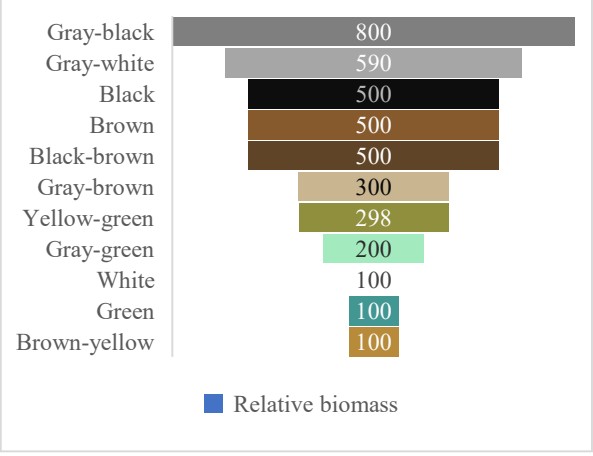

**Fig. 17** Relative biomass of biomes with different colors on the marble surface of the Hall of Prayer for Good Harvests at the Temple of Heaven in Beijing, China.

3.4 Relative biomass of communities' composition in different colored biological communities on rock surfaces in the study area

The relative biomass of different visual color biotic communities on the rock surface in the study area is shown (Fig. 18). An analysis of the main population compositions of these biological communities is presented (Fig. 19). The colors of biological communities on rock surfaces in the study area are primarily composed of black, brown, gray, green, and yellow, as well as combinations of these colors (gray-black, gray-white, black-brown, gray-brown, yellow-green, gray-green, and brown-yellow). The correlation between color combinations and population composition is not very apparent, which also indicates that determining microscopic color (population composition) through visual color is a complex and difficult task. Nevertheless, some patterns can be observed: Species like *Scytonema* sp.1, *Myxosarcina* sp., *Asterocapsa* sp., *Gomphosphaeria* sp., and *Gloeocapsa* sp.2 tend to make the community color darker, presenting as black, brown, gray, or combinations of these;

The parts that have a visual color of white are minerals, not biological organisms, under microscopic
observation; the areas with a visual color of green (mainly referring to the characteristic blue-green
of cyanobacteria) are mineral particles and *Chroococcus* sp.1; the areas with a visual color of
yellow-green are mainly mosses; the areas with a visual color of brown-yellow are mainly *Nostoc*
sp. etc.

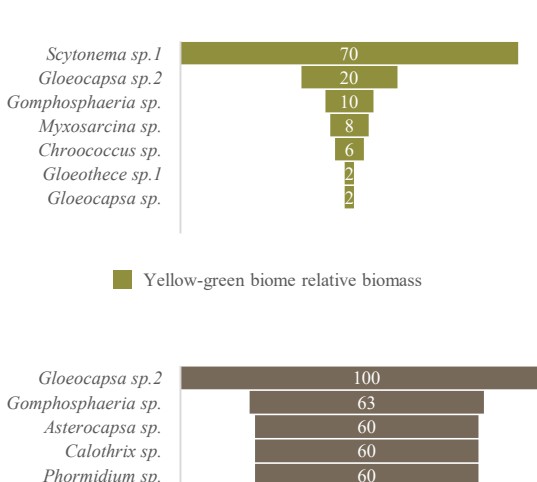

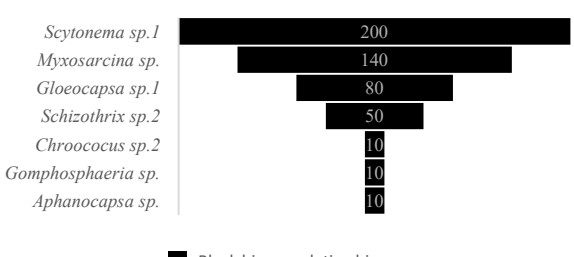

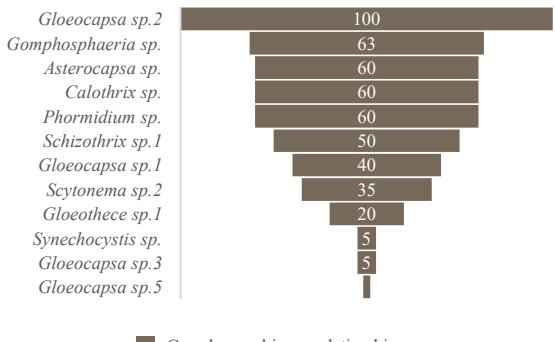

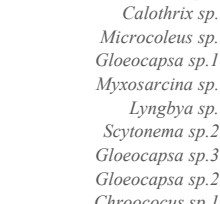

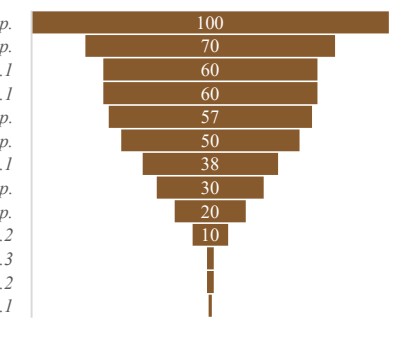

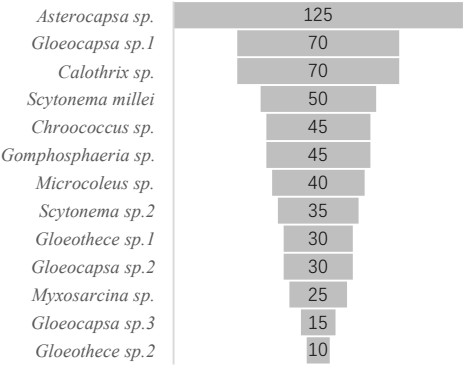

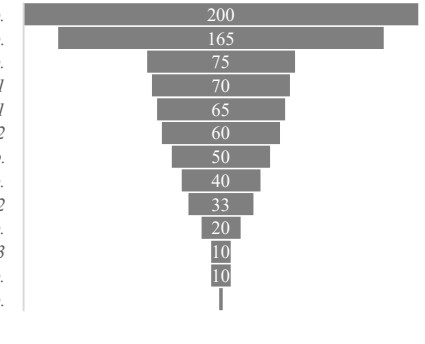

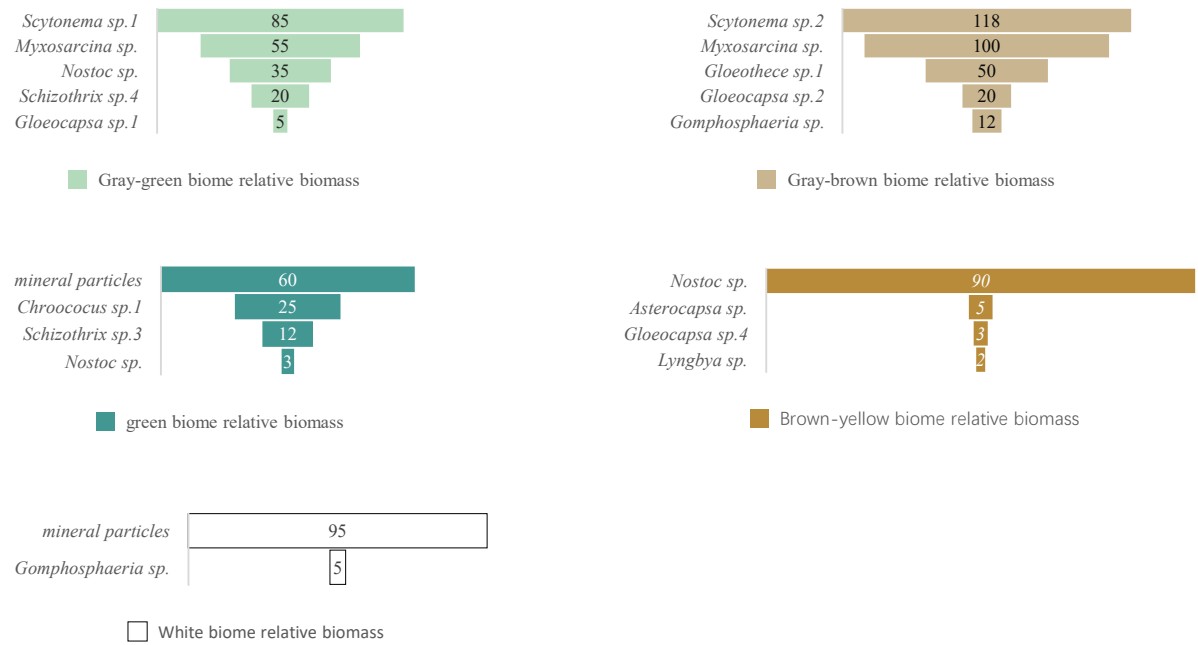

**Fig. 18.** Relative biomass of community composition of different colors on marble surface of the Hall of Prayer for Good Harvests at the Temple of Heaven in Beijing, China.

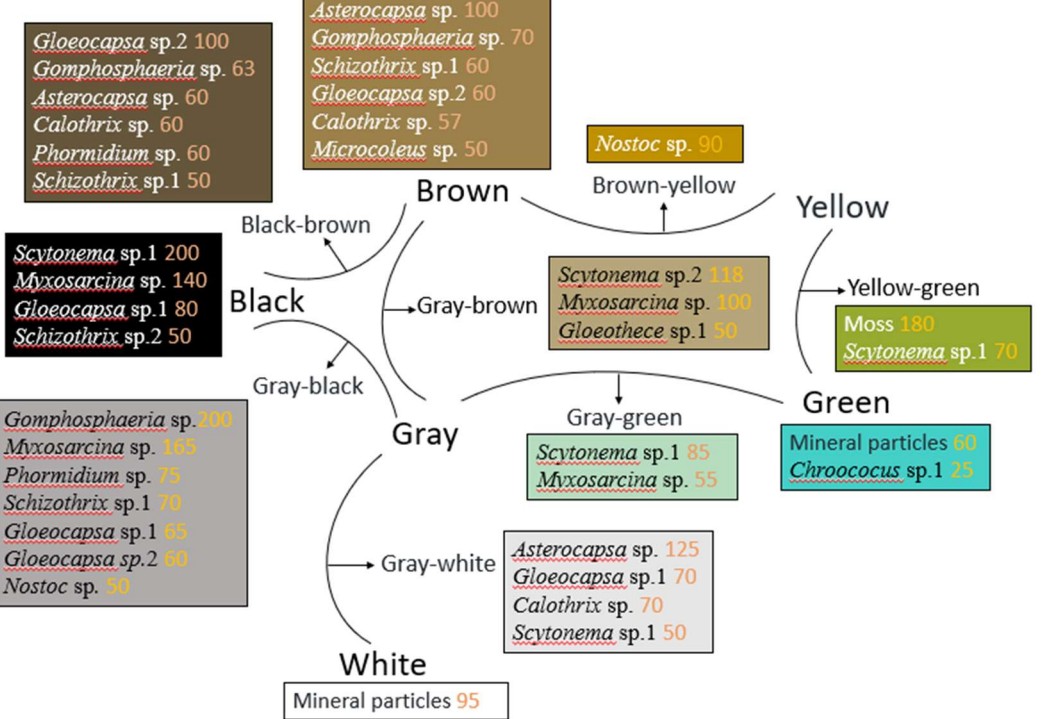

**Fig. 19.** Analysis of main population composition of different color biomes on marble surface of the Hall of Prayer for Good Harvests at the Temple of Heaven in Beijing, China.

3.5 Relative biomass of communities composition in different morphological biological
communities on rock surfaces in the study area
The biological communities on rock surfaces in the study area exhibit different morphologies,
including membranous, hairy, carpet-like, leathery, shell-like, and powdery layers. Their relative
biomass of population composition is shown (Fig. 20). A diagrammatic explanation of the formation
of these community morphologies is presented (Fig. 21).

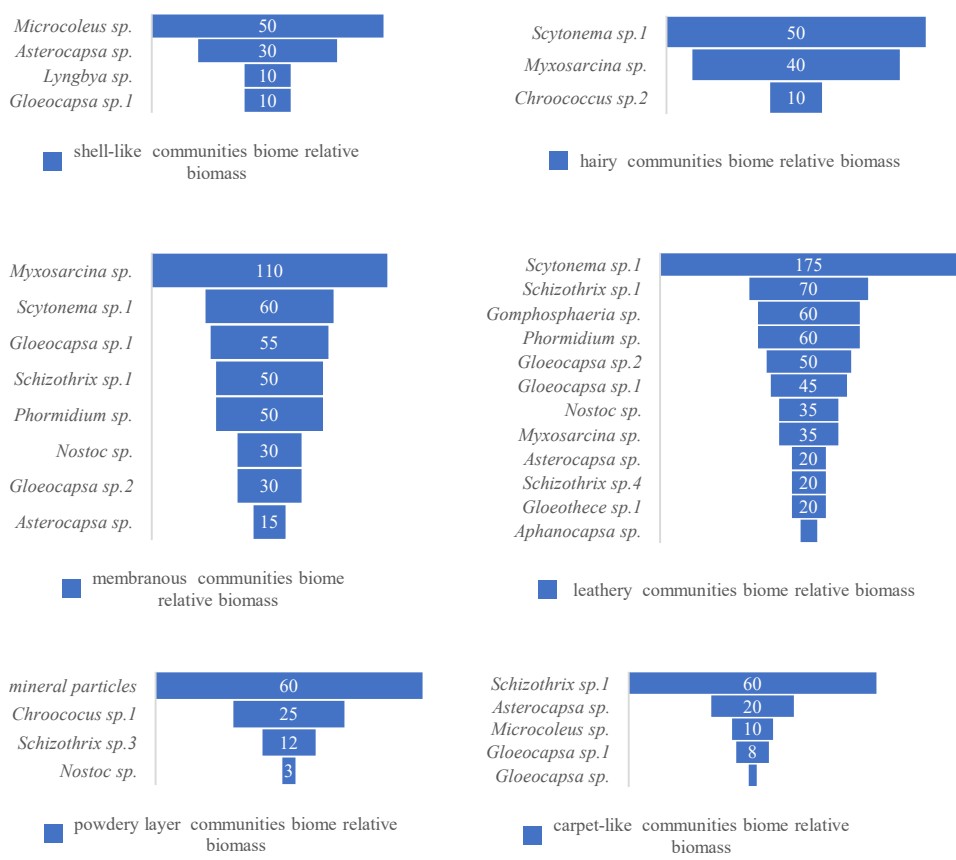

**Fig. 20.** Relative biomass of different forms of community on marble surface of the Hall of Prayer for Good

Harvests at the Temple of Heaven in Beijing, China.


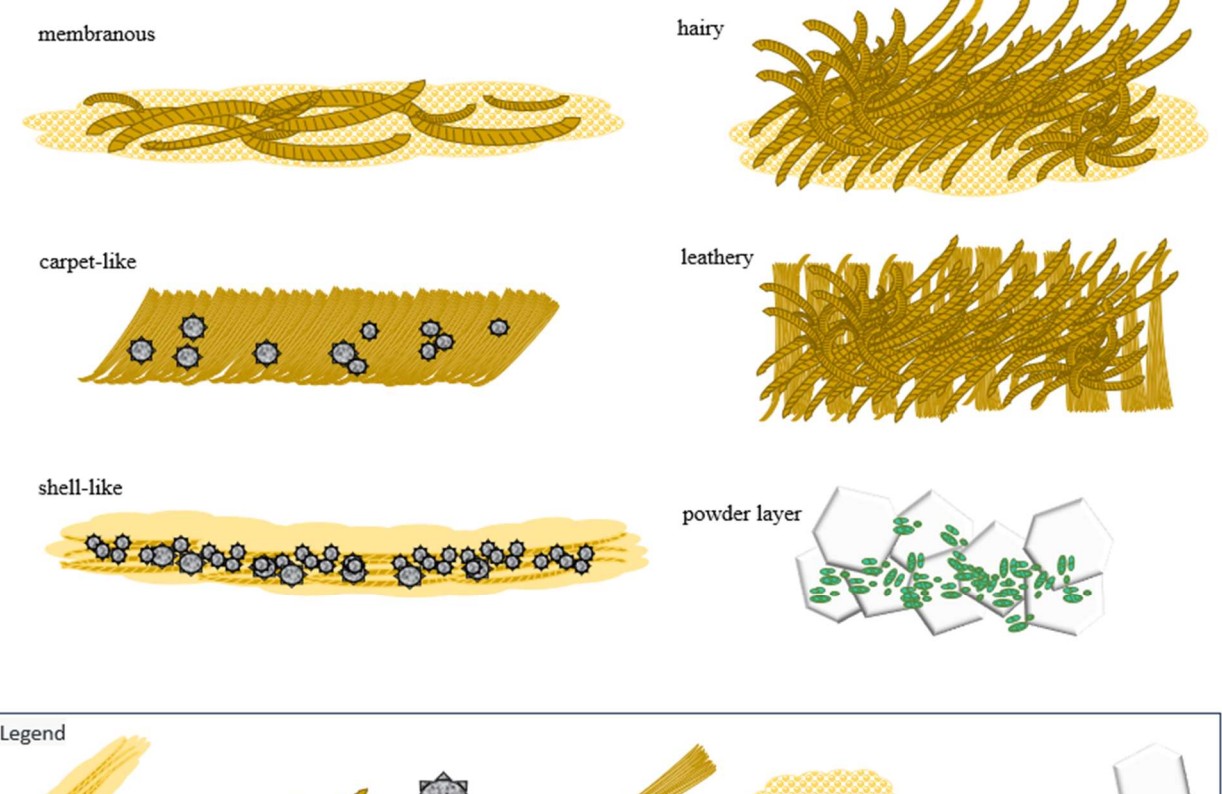

membranous

hairy

carpet-like

leathery

shell-like

powder layer

Legend

*Microcoleus* sp.      *Scytonema* sp.1      *Asterocapsa* sp.      *Schizothrix* sp.1      *Myxosarcina* sp.      *Chroococus* sp.1      mineral particles


**Fig. 21** Morphological genesis diagram of different communities on marble surface of the Hall of Prayer for Good
Harvests at the Temple of Heaven in Beijing, China.

The dominant species in the membranous biological communities are mainly *Myxosarcina* sp.
and *Scytonema* sp.1. The former accounts for a relative biomass of 110, while the latter accounts for
60(Fig. 17). *Myxosarcina* sp. is spherical cyanobacteria (Fig.4a). It has thick individual sheaths,
forming a dense colonial mucilage. *Scytonema* sp.1 grows interspersed within, forming a
membrane-like community (Fig.21). When the relative biomass of *Scytonema* sp.1 in the
community exceeds that of *Myxosarcina* sp., it forms a hairy community (Fig.21). The dominant
species in the carpet-like communities are mainly *Schizothrix* sp.1 and *Asterocapsa* sp. The former
accounts for a relative biomass of 60, while the latter accounts for 20 (Fig.20). *Schizothrix* sp.1
grows densely together, forming a carpet-like structure (Fig. 21). The dominant species in the
leathery biological communities are mainly *Scytonema* sp.1 and *Schizothrix* sp.1. The former
accounts for a relative biomass of 175, while the latter accounts for 70 (Fig. 20). *Scytonema* sp.1

intertwines, with *Schizothrix* sp.1 interspersed within (Fig. 21). The dominant species in the shell-like biological communities are mainly *Microcoleus* sp. and *Asterocapsa* sp. The former accounts for a relative biomass of 50, while the latter accounts for 30 (Fig.20). *Microcoleus* sp. has well-developed sheaths, with multiple algal filaments inside each sheath. The sheaths of multiple *Microcoleus* sp. aggregate to form a mucilaginous layer, with *Asterocapsa* sp.1 dispersed within. When the mucilaginous layer dries, it cracks into numerous small pieces. The edges of each piece detach from the rock surface and curl up, forming a shell-like structure(Fig. 21 and Fig. 22). The powder layer is a severely weathered surface (Fig. 13d). Under microscopic observation, it mainly consists of mineral particles and *Chroococus* sp.1, with the former accounting for 60 and the latter for 20 of the relative quantity (Fig. 20). *Chroococus* sp.1 is distributed on the surface and in the crevices of mineral particles (Fig. 14a and b). The color of the community appears as a mixture of the green color of *Chroococus* sp.1 (or the blue-green color characteristic of cyanobacteria) and the white color of mineral particles.

3.6 Bioweathering on Rock Surfaces in the Study Area

The growth distribution of aerophytic organisms on rock surfaces in the study area is closely related to the surface smoothness and texture of marble (Table 2). If the marble surface is uneven or has a non-uniform texture, the aerophytic organisms' communities will be distributed in a spotted pattern (Fig. 22a). Dissolution forms solution pits and cavities (Fig. 22b), which further expand into solution basins (Fig. 22c, d). If the marble surface has linear textures or non-uniform texture with joint stripes, the aerophytic organisms' communities will be distributed in a linear pattern (Fig. 22e). Dissolution forms solution marks and grooves (Fig.22f), which further expand into solution channels (Fig. 22g). If the marble surface is smooth and has a uniform texture, the aerophytic organisms' communities will be distributed in a planar pattern (Fig. 22h). Dissolution forms a weathering layer or spalling layer (Fig. 22i).

**Table 2**

Characteristics of the Marble Surface of the Hall of Prayer for Good Harvests in the Temple of Heaven, Beijing, China, and the Process of Biological Erosion on Its Surface.

| Marble Characteristics | Biological Community Distribution | Resulting Dissolution Forms | Development Process |
|---|---|---|---|
| Uneven surface or non-uniform texture | Spotted distribution | Solution pits, cavities, and basins | |
| Surface with linear textures or non-uniform texture with joint stripes | Linear distribution | Solution marks, grooves, and channels | ↓ |
| Smooth surface with uniform texture | Areal distribution | Weathering layer, spalling layer | |





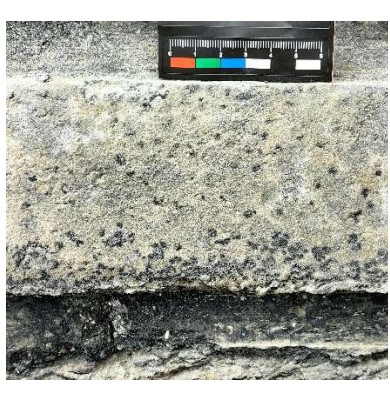

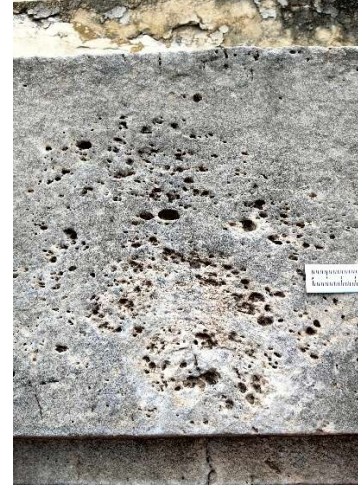

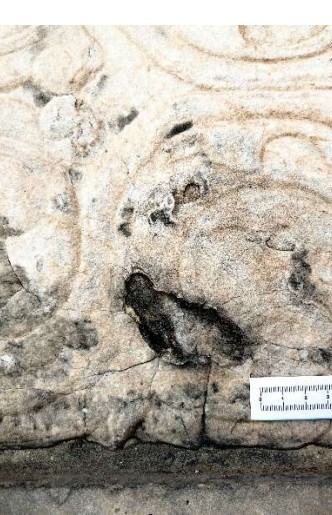

a. biological community point distribution

b. Solution pores and solution cavities

c. Sinkhole

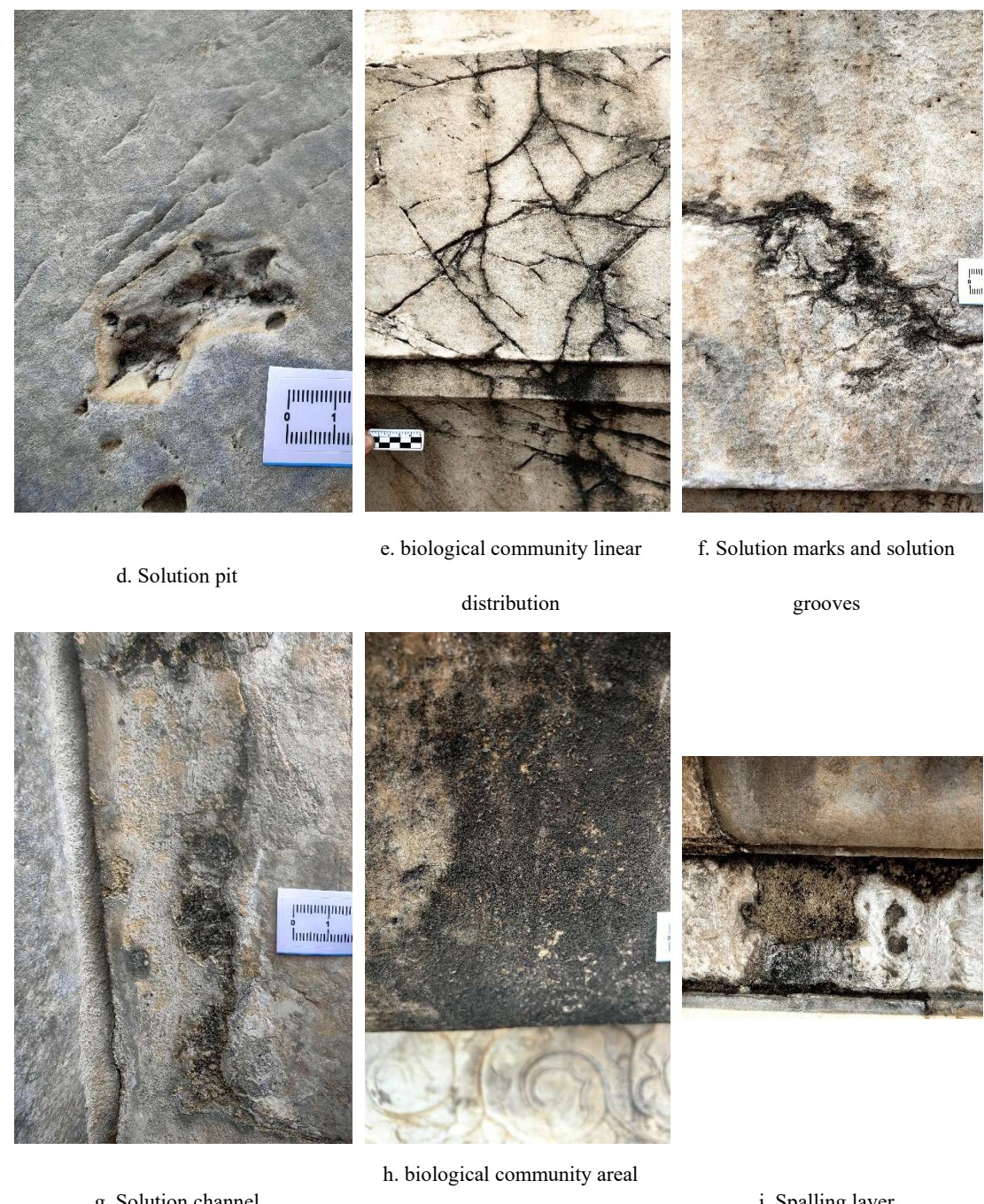

d. Solution pit      e. biological community linear distribution      f. Solution marks and solution grooves

g. Solution channel      h. biological community areal distribution      i. Spalling layer

**Fig. 22** Bioweathering forms on the marble surface of the Hall of Prayer for Good Harvests in the Temple of Heaven, Beijing, China.

The spotted distribution of biological communities gradually expands into linear distribution, and then into areal distribution. Solution pits, basins, and cavities also further enlarge their dissolution forms, developing into solution marks, grooves, and channels. For example, in the study

area, the weathering process of white marble "Cloud Chi Head" begins with the accumulation and
growth of organisms in the low-lying areas of the cloud patterns (Figures 23a, b).. These areas retain
more moisture, so they are the first to undergo bioweathering, forming deeper solution cavities and
channels. The communities then gradually spread to the surrounding areas, developing into linear
distributions, and then areal distributions, leading to flaking of the rock surface (Fig. 23c). This
partially destroys the pattern structure, further expanding the area and depth of dissolution, forming
a loose powder layer (Fig. 13a, d; Fig. 14a, b; Fig. 16b, c; Fig. 23d, e).

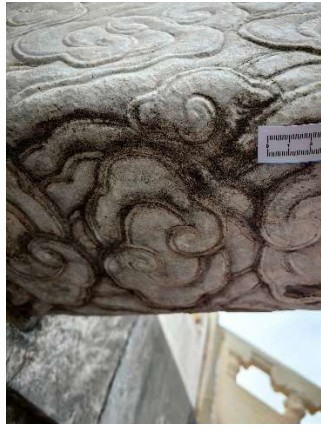

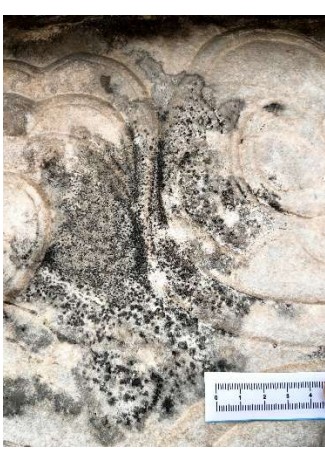

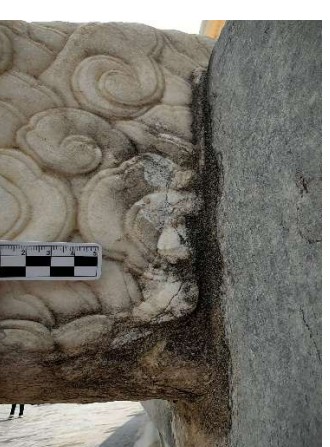

a. The organisms gather and grow in the low-lying areas of the Cloud Chi Head ornamentation.

b. The organisms gather and grow in the low-lying areas of the Cloud Chi Head ornamentation.

c. The surface of the Cloud Chi Head is flaking off in patches.

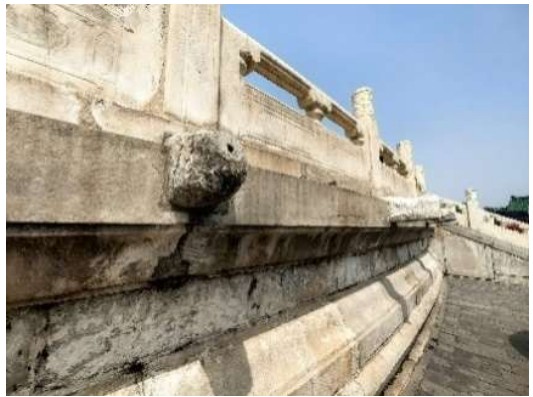

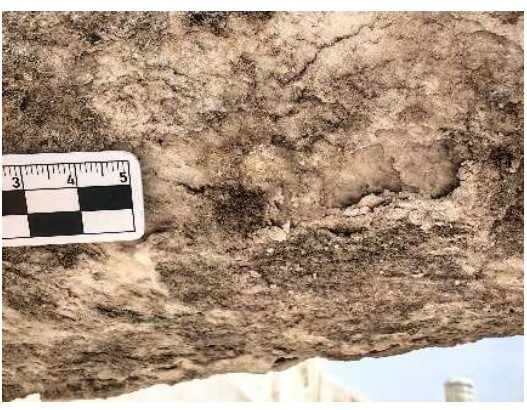

d. The Cloud Chi Heads have even weathered away completely.

e. A loose, powdery layer has formed on the surface of the Cloud Chi Heads, with a large amount of cyanobacteria growing inside.

**Fig. 23.** Bioweathering process of the Cloud Chi Head on the Hall of Prayer for Good Harvests in the Temple of
Heaven, Beijing,China.

4 Discussion
(1) This study focuses on the cyanobacterial and bryophyte communities that can be observed
using biological microscopy. The current scope of the research has not yet covered other microbial
groups. To determine whether other bacterial groups exist on the surface of stone cultural relics and
to understand their ecological functions, further systematic verification through subsequent studies
is still needed. At the methodological level for the classification and identification of cyanobacteria,
traditional morphological identification, although it may lead to taxonomic deviations at the genus
and species levels, molecular biology methods also face technical bottlenecks. For special samples
like biofilms on stone cultural relics, molecular testing typically requires microbial samples with a
high purity of more than 0.2 grams. However, in actual sampling, due to restrictions on cultural relic
protection, sometimes only trace amounts of less than 0.01 grams can be obtained. While such low
sample quantities are sufficient for morphological identification under a biological microscope, they
pose significant challenges for molecular biology methods. Low DNA extraction efficiency and
significant amplification bias from such small samples can result in decreased taxonomic resolution.
Furthermore, there has been long-standing controversy in the taxonomy of cyanobacteria. The
conflict between traditional morphological classification and molecular systematics has led to a
dynamic revision of the taxonomic framework. This instability makes it difficult to match taxonomic
information when annotating environmental samples using 16S rRNA gene sequence databases
(Lefler, et al., 2023). Future research should aim to construct a multidimensional identification
system, integrating microscopic observation, culturomics, and metagenomics, to gradually establish
classification standards and databases suitable for the study of microorganisms in cultural heritage.
This will be an important direction for the development of methodologies in this field.
(2) The differential weathering characteristics of the Cloud Chi Heads on the Hall of Prayer for
Good Harvests, as well as the directional differences in the spatial distribution of organisms on the
rock surface, show significant consistency. This correspondence confirms the scientific validity of
the visual analysis method based on the relative volume and the relative volume percentage
determined by microscopic observation. This method, through the analysis of micro-scale biotic
community features, can effectively reflect the differences in weathering processes in the macro-
environment, providing an important reference for establishing the correlation between micro-
observation indicators and macro-environmental factors.

(3) The bioweathering process of the marble at the Hall of Prayer for Good Harvests in the

Temple of Heaven is controlled by both macro-hydrological dynamics and micro-surface
topography: On a macro scale, in areas with low flow during heavy rain (raised areas), water quickly
drains away, resulting in sparse biofilms and weak bioweathering. In high-flow areas during heavy
rain (water-collecting grooves), the extended water retention time leads to the formation of "ink
bands" rich in cyanobacteria, resulting in strong bioweathering. On a micro scale, the micro-
topographic features of the rock determine the colonization patterns of organisms by regulating local
hydrological conditions—irregular rough surfaces induce point-like biological aggregation due to
discrete water films, leading to the development of solution pores and pits; linear decorations or
joint surfaces promote linear biological expansion due to directional water storage, forming solution
marks and grooves; smooth and dense surfaces support planar biological growth due to uniform
water film coverage, ultimately leading to the overall peeling of the weathered layer. This coupled
mechanism reveals that, in addition to the different sunlight exposure on the rock surface caused by
orientation, the synergistic regulation of spatiotemporal water distribution and rock surface
characteristics is also an important reason for the different distribution of biological communities
on stone cultural relics. Some studies also suggest that the type of stone, its position on the building,
and the surface roughness of the stone greatly influence biological growth (Korkanç and Savran,
2015). Some organisms (such as cyanobacteria and lichens) also bore into the marble, forming a
hard, black, porous layer (Golubić, et al., 2015). The biological black crust on marble is often
attributed to physical and inorganic chemical causes such as dust, which needs to be taken seriously.

(4) The connections and issues between different research levels, methods, and results in this

paper.

Connections and issues between different research levels, methods, and results in this paper.

This paper studies the aerophytic organisms on rock surfaces in the research area in terms of
biological community population composition, community color and morphology, and community
distribution characteristics (Table 3). The spotted, linear, and planar distributions of biological
communities on rock surfaces in the study area are composed of many microcommunities. These
microcommunities exhibit different morphologies, including membranous, hairy, carpet-like,

leathery, shell-like, and powder layers. Spotted, linear, and areal distributions of biological communities may be composed of one type of microcommunity or multiple types. Microcommunities are further composed of multiple populations, and a population consists of multiple individual organisms of the same species.

Community distribution characteristics are observed with the naked eye, without magnification. Community color and shape are observed through stereomicroscopes and the naked eye, magnifying objects 8-56 times (or no magnification if observed with the naked eye). Biological community population composition is identified through biological microscope observation, magnifying objects 40-1000 times. This represents three stages of research with increasing magnification of the research object: 1) Distribution area; 2) Community; 3) Population. Research at each stage is relatively easy to conduct, but the connections between stages are challenging and represent a key focus of this paper. For example, to accurately correlate different colored and shaped communities with their precise population compositions (i.e., connecting the community stage with the population stage) requires statistical analysis of numerous specimens to improve accuracy. Additionally, for outdoor observations of communities, which involve the transition between the distribution area stage and the community stage, the primary method is still visual observation with the naked eye. Only a small number of observations are conducted using stereomicroscopes because detailed stereomicroscopic observations that require photography must be done indoors. Sampling of cultural relics in scenic areas is extremely limited and must be carried out without damaging the relics. To address this issue, one approach is to enhance the performance of observation equipment to allow for in situ biological community observations outdoors without sampling, or to perform minimal sampling.

**Table 3**

Analysis of Research Levels in the Study on aerophytic organisms on marble of the Hall of Prayer for Good Harvests in the Temple of Heaven, Beijing, China.

| Research Level | Distribution Area | Community | Population |
| --- | --- | --- | --- |
| Observation Method | Naked eye | Stereomicroscope, naked eye | Biological microscope |
| Magnification | 0 | 8-56, 0 | 40-1000 |

| | | 11 colors: gray-black, gray-white, black, brown, black-brown, gray-brown, yellow-green, gray-green, white, green, brown-yellow | 30 genera and species: *Myxosarcina* sp., *Gomphosphaeria* sp., *Asterocapsa* sp. and so on (Fig.5) |
| Classification | 3 distribution characteristics (point, linear, and areal distribution) | | |
| | | 6 morphologies: membranous, hairy, carpet-like, leathery, shell-like, and powder layer | |
| Composition | Composed of multiple communities | Composed of multiple populations | Composed of multiple individuals of a single species |

(5) In this study, the weathering intensity of the Hall of Prayer for Good Harvests in the Temple of Heaven is found to be south-facing > west-facing > east-facing > north-facing. Additionally, the metabolic activity of the southeastern microbial community on the marble of Florence Cathedral is higher than that of the northwestern community (Checcucci, et al., 2022). This indicates that the weathering of stone cultural relics exhibits directional differences, and these directional differences vary in different climate zones. When studying the microenvironment of rock surfaces, temperature is relatively easy to measure, but humidity is difficult to measure accurately due to the significant influence of wind disturbances, which can lead to measurement failures. Therefore, more effective methods are needed to address this issue. Another comparison can be made between the marble relics of the Cathedral of Santa Maria del Fiore in Florence, Italy, and the Speranza statue (the Cathedral of Santa Maria del Fiore was completed in 1887 and is 135 years old; the Speranza statue was built in 1863 and is 158 years old). The growth of black biofilm on these structures is significantly more extensive and faster than that on the Hall of Prayer for Good Harvests at the Temple of Heaven in Beijing (built in 1420 and 605 years old). The primary reason for this difference is the climate. Florence has a Mediterranean climate with high rainfall (about 850 mm) (Venturi et al., 2020), while Beijing has a temperate monsoon climate with low annual rainfall

(During the period from 2009 to 2024, the multi-year average annual total rainfall was 610 mm,
according to data from the National Meteorological Science Data Center Website.). Therefore, water
is the primary factor determining the growth rate and distribution area of the black biofilm on marble.
Additionally, the different physical properties of marble in the two locations should also be
considered.

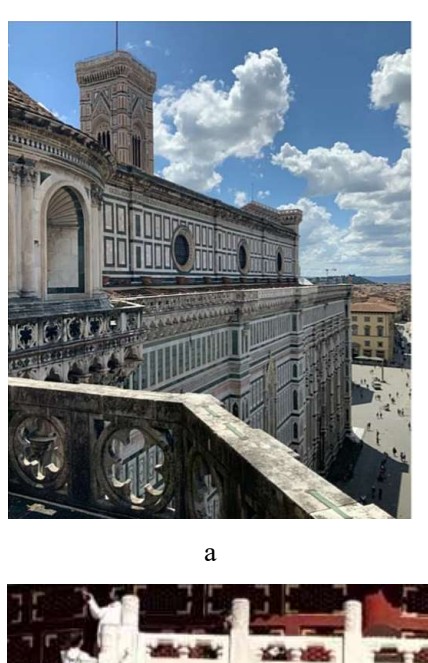 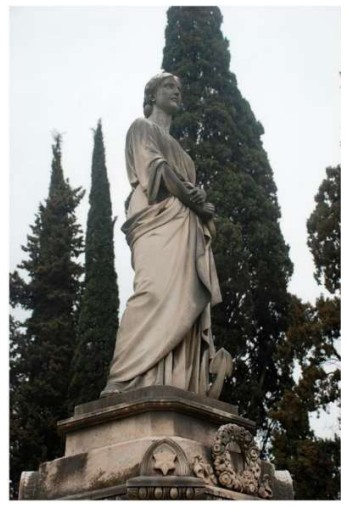

a                         b

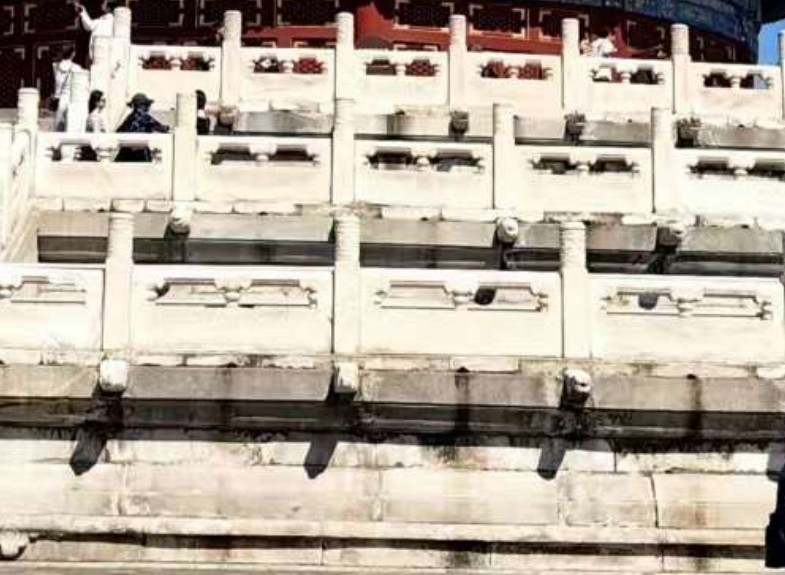

c

Fig. 24: Comparison of Black Biofilm Growth on Marble Relics in Florence and Beijing

a. Cathedral of Santa Maria del Fiore in Florence (façade completed in 1887, 138 years old, using white, green, and pink marble), with a
large area covered by black biofilm (Santo, 2023);
b. Speranza statue in Florence (built in 1863, 158 years old, using white marble), with a large area covered by black biofilm (Mascalchi,

2018);

c. The Hall of Prayer for Good Harvests at the Temple of Heaven in Beijing (built in 1420, 605 years old, using white and bluish-white marble), with a small area covered by black biofilm, mainly distributed in areas with high runoff from sudden rain.

(6) The species in the study area, such as *Scytonema* sp.2, are also common aerophytic cyanobacteria found on limestone surfaces (Tian, et al., 2002; Tian, et al., 2003; Tian, et al., 2004). They prefer calcareous environments, are drought-resistant, grow slowly, and have extremely strong vitality. The mechanism of dissolve rocks primarily involves the biological need to obtain inorganic nutrients such as calcium and magnesium ions from the rock. Aerophytic organisms can secrete organic acids, which release calcium and magnesium ions from the rock, providing the inorganic nutrients necessary for their growth and development. Through this acid dissolution process, aerophytic organisms can "eat away" at the rock, forming small hemispherical dissolution pits. This process damages the surface structure of the rock, leading to the formation of an underlying weathering layer (Tian, et al., 2004). In addition, various forms of cyanobacterial communities in extremely arid environments, such as on rocks, develop thicker exopolysaccharide (EPS) sheaths to retain intracellular water. The EPS sheath undergoes contraction and expansion in response to changes in weather conditions, accelerating the disintegration of rock particles on the surface of rocks. This process is very similar to microbial weathering in the Atacama Desert (Jung, et al., 2020). Both processes involve the swelling of the EPS due to water absorption, leading to the deformation of the biofilm and the detachment of the rock surface at the community scale. This results in the expansion of patchy weathering into a more extensive weathering layer (such as the Atacama terrestrial protopedon or the powdery layer at the Temple of Heaven) at the landscape scale. The mechanism involves the tensile stress generated by the swelling of the EPS exceeding the local tensile strength of the rock, initiating cracks (such as grain boundary cracking in the Atacama and mineral particle detachment at the Temple of Heaven). These cracks provide pathways for chemical and biological erosion, leading to an expanded pore/crack network, increased water retention time, enhanced biological activity, and further swelling, creating a self-reinforcing weathering loop. It is clear that the swelling effect plays a crucial role as a "physical engine" in microbial bioweathering. Future research should focus on cross-scale mechanical modeling: scaling up the swelling force of microbial EPS (at the nN level) to the point of rock fracture (at the MPa level) to reveal the mechanisms of scale transition; quantifying the impact of changes in fog/rain patterns under global warming on the frequency of biological swelling, to warn of accelerated weathering risks; and

recognizing that swelling not only acts as a "trigger" for rock destruction but also serves as a key link between biological activity and surface processes. Its universality across different environments provides a new perspective for understanding the evolution of the Earth's critical zone.

5 Conclusion

(1) The most dominant species on marble surfaces in the study area is *Myxosarcina* sp., followed by *Gomphosphaeria* sp., *Asterocapsa* sp.1, *Gloeocapsa* sp.1, and *Scytonema* sp.1. These aerobic cyanobacteria prefer calcareous environments, are drought-tolerant, slow-growing, and extremely resilient.

(2) The biological population composition on marble surfaces facing different directions of the Hall of Prayer for Good Harvests in the Temple of Heaven varies due to differences in sunlight exposure. The east-facing side, warm and humid, mainly hosts small filamentous and spherical cyanobacteria such as *Scytonema* sp.2 and *Gomphosphaeria* sp. The west-facing side, hot and humid, primarily features *Scytonema* sp.1 and mosses, with *Scytonema* sp.1 being small filamentous cyanobacteria. The north-facing side, cold and humid, mainly supports spherical cyanobacteria like *Myxosarcina* sp. and *Gomphosphaeria* sp. The south-facing side, hot and dry, primarily hosts small or large filamentous cyanobacteria such as *Scytonema* sp.1 and *Nostoc* sp.. The observed weathering intensity in different directions is: south > west > east > north, which is entirely consistent with the varying degrees of weathering reflected by the Cloud Chi Heads in each direction. This indicates that the visual analysis method based on the relative volume and relative volume percentage of species, as determined by microscopic observation and statistical analysis, is scientifically valid.

(3) Rock surface biological communities in the study area display various colors, with gray-black being the most common, followed by gray-white, black, brown, and brown-black. Gray-black communities are mainly composed of *Myxosarcina* sp. and *Gomphosphaeria* sp.

(4) Rock surface biological communities in the study area exhibit different morphologies, including membranous, hairy, carpet-like, leathery, shell-like, and powder layers. Different morphologies correspond to different population compositions.

(5) In addition to sunlight exposure, the growth of aerial organism on the rock surfaces in the study area is also controlled by macro-hydrological dynamics and micro-surface topography. On a

macro scale, in areas with low flow during heavy rain, the biofilm is sparse, and the bioweathering effect is weak. In areas with high-flow areas during heavy rain, "ink bands" rich in cyanobacteria form, leading to strong bioweathering. On a micro scale, the microtopographic features of the rock regulate local hydrological conditions, determining the colonization patterns of the organisms: On uneven or heterogeneous marble surfaces, aerial organism communities are distributed in patches, leading to the formation of solution pores, cavities, and pits; On marble surfaces with linear patterns or heterogeneous textures with joint lines, aerial organism communities are distributed in linear patterns, leading to the formation of solution marks, grooves, and channels. On flat and homogeneous marble surfaces, aerial organism communities are distributed in a planar pattern, leading to the formation of weathering layers or spalling layers. The thicker exopolysaccharide (EPS) sheath of aerophytic cyanobacteria can undergo contraction and expansion, thereby accelerating the disintegration of rock particles on the surface of rocks. Preventing or reducing the growth of aerial organism is key to slowing down the bioweathering process of the marble at the Hall of Prayer for Good Harvests in the Temple of Heaven.

**Author contributions**

YT completed all the work on the paper, including sampling, photography, experimental data analysis, charting, drawing, and writing the paper, among other tasks.

**Competing interests**

The author has declared that there are no competing interests.

**Acknowledgement**

This study was supported by National Science Foundation of China (Grant No. 40872197) and the Development Fund of China University of Geosciences (Beijing) (Grant No. F02114).

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
