# Peer review of "Study on the Biological Communities and Bioweathering of the"

_EGUsphere, 2024_

## Referee Comment (RC2)

This is an interesting paper which describes epilithic phototrophic biofilms on a cultural heritage monument in Bejing, the Temple of Heaven. These biofilms have the potential for bioweathering, i.e. the degradation/dissolution of lithic structures such as marble.

Although the paper is generally well written I have the following critical comments:

As the paper is focused on cyanobacterial biofilms avoid the term "alga/e". Cyanobacteria are prokaryotes and NOT eukaryotic algae!

introduction should be more condensed, in particular the historical aspects of the Temple of Heaven since the focus is bioweathering by phototrophic biofilms. In addition, although I acknowledge many Chinese colleagues working on bioweathering/biodeterioation problems, more colleagues from Europe should be cited, as they were studying such problems already in the 90ies.

Figure 2 – data set stops in 2011 – I think more recent data should be included since particularly during the last decade many meteorological changes due to global warming can be documented

Methods – I deeply doubt species identification, because most of the cited literature is outdated and because cyanobacteria are extremely difficult or even impossible to identify based on morphology only. These days, you need molecular-genetic data (16S rDNA or other specific markers) to prove identity. In addition, taxonomic assignments ("species names") regularly change, and hence a data base (e.g. AlgaeBase) has to be consulted before submission of a manuscript. Perhaps to overcome the taxonomic problems the author might consider to always stick to the genus level only, and NOT to mention species.

3.1 change "population distribution" to "community distribution" (population represents genotypes of the same species!), and throughout the whole manuscript.

Line 190/191 – do not discuss your data in the results

3.2 change "Biological Population Distribution" to "community distribution" and throughout the whole manuscript

Line 213 – do not discuss your data in the results

Fig. 5a/b – delete Chinese letters

Fig. 7 e/f, 9f are of bad quality, and taxa can not be identified!

Table 1 and 2 – although environmental data are very important for such a study, there are real data missing. Less or more sunlight does not tell the reader anything - real data instead would help! What was the temperature in the sun or at shaded sites?

Fig. 18 – very nice!

In the discussion I miss more European studies. See also the recent papers of Patrick Jung on bioweathering mechanisms of rocks in the Atacama desert by cyanobacteria and cyanolichens!

My general recommendation is as follows:

The topic is interesting and addresses the scope of the journal. Biological vocabulary/wording has to be carefully modified (see my comment on population). Focus on cyanobacteria.

Identification/taxonomy is highly questionable (see my comments). The macroscopic description of the biofilms is well done. Microscopic images of the cyanobacteria are not always of appropriate quality. Real environmental data should be provided. English grammar and expression is fine. I think the whole manuscript could be condensed (e.g. historical aspects in the introduction, number of figures etc.).

I recommend a major revision

---

## Author Comment (AC2)

Dear Reviewer,

Thank you very much for your detailed review of our manuscript and for the valuable comments you have provided. Your feedback is crucial for improving the quality of our paper. Below are our specific responses and planned revisions to each of your comments:

**Content of the Paper**

You mentioned that this paper describes the photosynthetic biofilms on the cultural heritage site of the Temple of Heaven in Beijing and points out the potential bio-weathering effects of these biofilms, i.e., the degradation and dissolution of marble and other rock structures. We fully agree with this assessment and appreciate your recognition.

**Critical Comments**

**Terminology**

You suggested avoiding the term "algae" and pointed out that cyanobacteria are prokaryotes, not eukaryotic algae. We fully agree with your view and have revised all mentions of "algae" in the manuscript to "cyanobacteria."

**Introduction**

You recommended that the introduction should be more concise, especially regarding the historical aspects of the Temple of Heaven. Since the focus of our paper is on the bio-weathering caused by photosynthetic biofilms, we will streamline the historical section to make it more compact and targeted.

**Citation of European Research**

You acknowledged the work of Chinese researchers and also suggested citing more European studies, particularly those that began researching bio-weathering in the 1990s. We understand your suggestion and plan to include more references to European research.

**Update of Chart Data**

You pointed out that the dataset in Figure 2 is up to 2011 and recommended including more recent data. We fully agree and plan to collect updated data, especially considering the significant meteorological changes due to global warming over the past decade. These data will make the paper more timely and scientific.

**Methodology and Species Identification**

You expressed doubts about the accuracy of species identification in the research methods and noted that many of the cited references are outdated, and identification based solely on morphology is difficult for cyanobacteria. You suggested using molecular genetic data (such as 16S rDNA or other specific markers) to determine species identity. This is part of the research I plan to conduct in the future and intend to publish in subsequent articles. For this paper, I will follow your advice and mention only the genus level without specifying the species. Additionally, I will consult relevant databases (such as the Algae Database) before submission to ensure the accuracy of taxonomic classification ("species names").

**Unification of Biological Community Terminology**

You suggested changing "population distribution" to "community distribution" because a population represents the genotype of the same species. We understand your point and will revise all mentions of "population distribution" in the manuscript to "community distribution."

Separation of Results and Discussion

You recommended avoiding discussion of data in the results section. We will strictly separate the results and discussion sections, ensuring that the results section contains only observations and data presentation, while the discussion section analyzes the significance of these data and interprets the results.

Chart Processing

You noted that Figure 5a/b contains Chinese characters and suggested removing them. We will follow this suggestion to make the charts more concise and understandable for international readers.

Image Quality

You mentioned that the quality of Figure 7 e/f and Figure 9f is poor, making it difficult to identify taxonomic units. We will reassess these images and, if possible, retake or obtain higher-quality images to ensure clear identification of taxonomic units.

Supplement of Table Data

You pointed out that Tables 1 and 2 lack actual environmental data. We recognize the importance of environmental data and the missing details, such as the inability to inform readers about specific temperature information due to the lack of light intensity data. We will supplement actual environmental data, such as specific temperatures in sunlit and shaded areas.

Discussion Section

You noted that the discussion section lacks more European studies and recommended referencing Patrick Jung's paper on the bio-weathering mechanisms of cyanobacteria and cyanolichens on rocks in the Atacama Desert. We will review and cite these European studies to make the discussion more comprehensive and in-depth.

Overall Revision Suggestions

You recommended streamlining the entire paper, mentioning more historical aspects in the introduction, and reducing the number of charts. We will follow these guidelines to make the paper more concise, clear, and targeted.

Once again, thank you for your professional comments and suggestions.

Best regards,

Tian Youping
China University of Geosciences (Beijing)
typ@cugb.edu.cn

---

## Author Comment (AC3)

Dear Reviewer,

Thank you very much for your detailed review of my manuscript and for the valuable comments you have provided. Your feedback is crucial for improving the quality of my paper. Below are my specific responses and planned revisions to each of your comments:

**Content of the Paper**

You mentioned that this paper describes the photosynthetic biofilms on the cultural heritage site of the Temple of Heaven in Beijing and points out the potential bio-weathering effects of these biofilms, i.e., the degradation and dissolution of marble and other rock structures. I fully agree with this assessment and appreciate your recognition.

**Critical Comments**

**Terminology**

You suggested avoiding the term "algae" and pointed out that cyanobacteria are prokaryotes, not eukaryotic algae. I fully agree with your view and have revised all mentions of "algae" in the manuscript to "cyanobacteria."

**Introduction**

You recommended that the introduction should be more concise, especially regarding the historical aspects of the Temple of Heaven. Since the focus of my paper is on the bio-weathering caused by photosynthetic biofilms, I will streamline the historical section to make it more compact and targeted.

**Citation of European Research**

You acknowledged the work of Chinese researchers and also suggested citing more European studies, particularly those that began researching bio-weathering in the 1990s. I understand your suggestion and plan to include more references to European research.

**Update of Chart Data**

You pointed out that the dataset in Figure 2 is up to 2011 and recommended including more recent data. I fully agree and plan to collect updated data, especially considering the significant meteorological changes due to global warming over the past decade. These data will make the paper more timely and scientific.

**Methodology and Species Identification**

You expressed doubts about the accuracy of species identification in the research methods and noted that many of the cited references are outdated, and identification based solely on morphology is difficult for cyanobacteria. You suggested using molecular genetic data (such as 16S rDNA or other specific markers) to determine species identity. This is part of the research I plan to conduct in the future and intend to publish in subsequent articles. For this paper, I will follow your advice and mention only the genus level without specifying the species.

**Unification of Biological Community Terminology**

You suggested changing "population distribution" to "community distribution" because a population represents the genotype of the same species. I understand your point and will revise all mentions of "population distribution" in the manuscript to "community distribution."

**Separation of Results and Discussion**

You recommended avoiding discussion of data in the results section. I will strictly separate the results and discussion sections, ensuring that the results section contains only observations and data presentation, while the discussion section analyzes the significance of these data and interprets the results.

**Chart Processing**

You noted that Figure 5a/b contains Chinese characters and suggested removing them. I will follow this suggestion to make the charts more concise and understandable for international readers.

**Image Quality**

You mentioned that the quality of Figure 7 e/f and Figure 9f is poor, making it difficult to identify taxonomic units. I will reassess these images and, if possible, retake or obtain higher-quality images to ensure clear identification of taxonomic units.

**Supplement of Table Data**

You pointed out that Tables 1 and 2 lack actual environmental data. I recognize the importance of environmental data and the missing details, such as the inability to inform readers about specific temperature information due to the lack of light intensity data. I will supplement actual environmental data, such as specific temperatures in sunlit and shaded areas.

**Discussion Section**

You noted that the discussion section lacks more European studies and recommended referencing Patrick Jung's paper on the bio-weathering mechanisms of cyanobacteria and cyanolichens on rocks in the Atacama Desert. I will review and cite these European studies to make the discussion more comprehensive and in-depth.

**Overall Revision Suggestions**

You recommended streamlining the entire paper, mentioning more historical aspects in the introduction, and reducing the number of charts. I will follow these guidelines to make the paper more concise, clear, and targeted.

Once again, thank you for your professional comments and suggestions.

Best regards,
Tian Youping
China University of Geosciences (Beijing)
typ@cugb.edu.cn

---

## Author Response (AR1)

**Dear Editor and Reviewers,**

Thank you for giving me the opportunity to submit my revised draft of the manuscript (ID: egusphere-2024-2758), entitled "Study on the Biological Communities and Bioweathering of Marble Surfaces at Temple of Heaven Park, Beijing, China". I sincerely thank the editor and all reviewers for their valuable feedback on my manuscript. Their insightful comments have significantly contributed to improving the quality of my work. In accordance with their suggestions, I have made the detailed corrections marked in red in the revised manuscript. Below, I address each of the issues you raised and describe the revisions I made to improve the manuscript.

Sincerely,
Youping Tian Ph.D
Professor of China University of Geosciences (Beijing)
E-mail: typ@cugb.edu.cn

**Response to Reviewer #1:**

1. The language should be revised to have a homogeneous scientific text. There are terms here and there which do not correspond to scientific language.

**Response:** Thanks for your suggestion. A comprehensive review of all the terminology used in the article has been conducted to ensure that all technical terms are appropriately used in this context. Uncommon terms are clearly defined to help readers better understand. Specifically, as follows:

(1) "Relative Volume" and "Relative Volume percentage" have been clearly defined and illustrated with a diagram (Fig. 4). **Lines 248-260**

(2) The term "aerophytic algae" has been changed to "aerial cyanobacteria" because "aerophytic" refers to aerial plants. After changing " blue-green algae" to "cyanobacteria," using "aerophytic" would be inappropriate.

2.Some edition (i.e. being concise) in the introduction must be done to keep the relevance in the story. Also elaborate in the scientific assertions made in the introduction.

**Response:** Thank you pointing these out. The introduction has been rewritten. In the introduction, the language has been simplified, and unnecessary redundant information has been removed to make the core content more focused and relevant. For the scientific assertions made in the introduction, more detailed background information, relevant theoretical support, and references are provided, appropriately explaining how these assertions lay the foundation for subsequent research or discussion. **Lines 47-158**

**Response to Reviewer #2:**

This is an interesting paper which describes epilithic phototrophic biofilms on a cultural heritagemonument in Bejing, the Temple of Heaven. These biofilms have the potential for bioweathering, i.e. the degradation/dissolution of lithic structures such as marble. Although the paper is generally well written I have the following critical comments:

**Response:** Thank you for giving me the opportunity to submit a revised draft of the manuscript. I sincerely appreciate you for your valuable feedback on my manuscript. Following your suggestions, I have made the detailed revisions outlined below.

    1. As the paper is focused on cyanobacterial biofilms avoid the term "alga/e". Cyanobacteria are prokaryotes and NOT eukaryotic algae!

**Response:** Thanks for your suggestion. All instances of "blue-green algae" in the paper have been changed to "cyanobacteria."

    2. introduction should be more condensed, in particular the historical aspects of the Temple of Heaven since the focus is bioweathering by phototrophic biofilms.

**Response:** Thank you for your comments. The historical background in the introduction has been streamlined. It has been reduced from **Lines 36-44** (a total of 9 lines) in the original manuscript to **Lines 49-54** (a total of 6 lines) in the revised manuscript.

    3. In addition, although I acknowledge many Chinese colleagues working on bioweathering/biodeterioation problems, more colleagues from Europe should be cited, as they were studying such problems already in the 90ies.

**Response:** Thank you for your valuable suggestions. European research findings have been added, involving the following literature. A comparative discussion with European studies has been included in the introduction (**Lines 69-126**) and discussion (**Lines 643-701**).

Checcucci, A, Borruso, L, Petrocchi, D, Perito, B.: Diversity and metabolic profile of the microbial communities inhabiting the darkened white marble of Florence Cathedral. International Biodeterioration & Biodegradation, 171- . https://doi.org/10.1016/j.ibiod.2022.105420, 2022

Gioventù, E., Lorenzi, P. F., Villa, F., Sorlini, C., Rizzi, M., Cagnini, A., Griffo, A., Cappitelli, F.: Comparing the bioremoval of black crusts on colored artistic lithotypes of the Cathedral of Florence with chemical and laser treatment. International Biodeterioration & Biodegradation, 65(6): 832-839, https://doi.org/10.1016/j.ibiod.2011.06.002, 2011

Gorbushina, A. Lyalikova, N. Vlasov, D.Y. Khiznyak, T.: Microbial communities on the monuments of Moscow and St. Petersburg: biodiversity and trophic relations. Microbiology, 71, 350-356, https://doi.org/10.1023/A:1015823232025, 2002

Isola, D., Zucconi, L., Onofri, S., Caneva, G., Selbmann, L.: Extremotolerant rock inhabiting black fungi from Italian monumental sites. Fungal Diversity 76, 75–96. https://doi.org/10.1007/s13225-015-0342-9, 2016

Jung, P., Baumann, K., Emrich, D., Springer, A., Felde, V. J. M. N. L., Dultz, S., Baum, C., Frank, M., Büdel, B., Leinweber, P.: Lichens Bite the Dust – A Bioweathering Scenario in the Atacama Desert. iScience, 23, 101647. https://doi.org/10.1016/j.isci.2020.101647, 2020

Leo, F., D., Antonelli, F., Pietrini, A., M., Ricci, S., Urzì, C.: Study of the euendolithic activity of black meristematic fungi isolated from a marble statue in the Quirinale Palace's Gardens in Rome, Italy. Facies, 65,18, https://doi.org/10.1007/s10347-019-0564-5, 2019

Marvasi, M., Donnarumma, F., Frandi, A., Mastromei, G., Sterflinger, K., Tiano, P., Perito, B.: Black microcolonial fungi as deteriogens of two famous marble statues in Florence, Italy,

International Biodeterioration & Biodegradation, Volume 68, 36-44, https://doi.org/10.1016/j.ibiod.2011.10.011, 2012

Monte, M Del, Sabbioni, C.: Chemical and bioweathering of an historical building: Reggio Emilia Cathedral. Science of The Total Environment, 50: 165-182, https://doi.org/10.1016/0048-9697(86)90358-X., 1986

Mascalchi, M., Osticioli, I., Cuzman, O. A., Mugnaini, S., Giamello, M., Siano, S.: Laser removal of biofilm from Carrara marble using 532 nm: The first validation study, Measurement, V. 130, 255-263, https://doi.org/10.1016/j.measurement.2018.08.012 , 2018

Moropoulou, A, Bisbikou, K, Torfs, K, Van Grieken, R, Zezza, F, Macri, F.: Origin and growth of weathering crusts on ancient marbles in industrial atmosphere. Atmospheric Environment, 32(6): 967-982, https://doi.org/10.1016/S1352-2310(97)00129-5, 1998

Pinna, D, Galeotti, M, Perito, B, Daly, G, Salvadori, B.: In situ long-term monitoring of recolonization by fungi and lichens after innovative and traditional conservative treatments of archaeological stones in Fiesole (Italy), International Biodeterioration & Biodegradation, Volume 132, 49-58, https://doi.org/10.1016/j.ibiod.2018.05.003, 2018

Santo, A., P., Agostini, B., Cuzman, O., A., Michelozzi, M., Salvatici, T., Perito, B.: Essential oils to contrast biodeterioration of the external marble of Florence Cathedral, Science of The Total Environment,V. 877, https://doi.org/10.1016/j.scitotenv.2023.162913,2023

Trovão, J., Portugal, A.: The impact of stone position and location on the microbiome of a marble statue. The Microbe, 2, 100040. https://doi.org/10.1016/j.microb.2024.100040, 2024

Venturi, S., Tassi, F., Cabassi, J., Gioli,B., Baronti, S., Vaselli, O., Caponi,C., Vagnoli,C. Picchi, G., Zaldei, A., Magi, F., Miglietta, F., Capecchiacci, F. : Seasonal and diurnal variations of greenhouse gases in Florence (Italy): Inferring sources and sinks from carbon isotopic ratios, Science of The Total Environment, Volume 698, https://doi.org/10.1016/j.scitotenv.2019.134245, 2020

4. Figure 2 – data set stops in 2011 – I think more recent data should be included since particularly during the last decade many meteorological changes due to global warming can be documented

**Response:** Thank you for raising this point. The data in the original Figure 2 has been re-plotted with the latest data in Figures 2 and 3, see **Lines 177-190**.

5. Methods – I deeply doubt species identification, because most of the cited literature is outdated and because cyanobacteria are extremely difficult or even impossible to identify based on morphology only. These days, you need molecular-genetic data (16S rDNA or other specific markers) to prove identity. In addition, taxonomic assignments ("species names") regularly change, and hence a data base (e.g. AlgaeBase) has to be consulted before submission of a manuscript. Perhaps to overcome the taxonomic problems the author might consider to always stick to the genus level only, and NOT to mention species.

**Response:** Thank you for your comments. I sincerely apologize for the omission of the most recent taxonomic references in the initial draft of my research. During the revision process, I have thoroughly reviewed and added the following key references:

Komarek, J., Anagnostidis, K.: Cyanoprokaryota 2. Teil: Oscillatoriales. Vol. 19(2), In: Büdel B., Gärtner G, Krienitz, L. and Schagerl, M. (eds.), Suesswasserflora von Mitteleuropa. Heidelberg, Germany: Springer Spektrum. 1-759, 2005.

Komarek, J., Büdel, B., Gärtner, G. Süßwasserflora von Mitteleuropa, Bd. 19/3: Cyanoprokaryota: 3. Teil / 3rd part: Heterocytous Genera. Springer Spektrum, 2013

All species names in the text have been changed to genus names, and the correspondence between the species names in the revised manuscript and the original manuscript is shown in the table below.

**Table   The correspondence between the species names in the revised manuscript and the original manuscript**

| the species names of the original manuscript | the species names of the revised manuscript | the species names of the original manuscript | the species names of the revised manuscript |
| --- | --- | --- | --- |
| *Myxosarcina* sp. | *Myxosarcina* sp. | *Chroococus membraninus* | *Chroococus* sp.1 |
| *Gomphosphaeria* sp. | *Gomphosphaeria* sp. | *Schizothrix telephoroides* | *Schizothrix* sp.2 |
| *Asterocapsa atrata* | *Asterocapsa* sp. | *Schizothrix delicatissima* | *Schizothrix* sp.3 |
| *Gloeocapsa crepidinum* | *Gloeocapsa sp.*1 | *Schizothrix* sp. | *Schizothrix* sp.4 |
| *Scytonema millei* | *Scytonema* sp.1 | *Lyngbya* sp. | *Lyngbya* sp. |
| *Scytonema bohneri* | *Scytonema* sp.2 | *Chroococcus* sp. | *Chroococus* sp.2 |
| *Gloeocapsa rupicola* | *Gloeocapsa* sp.2 | *Gloeothece fusco-lutea* | *Gloeothece* sp.2 |
| *Calothrix* sp. | *Calothrix* sp. | *Chroococcus lithophilus* | *Chroococus* sp.3 |
| *Nostoc calcicole* | *Nostoc* sp. | *Aphanocapsa muscicola* | *Aphanocapsa* sp. |
| *Gloeothece rupestris* | *Gloeothece* sp.1 | *Synechocystis* sp. | *Synechocystis* sp. |
| *Phormidium* sp. | *Phormidium* sp. | *Gloeocapsa* sp. | *Gloeocapsa* sp.3 |
| *Schizothrix fasciculata* | *Schizothrix* sp.1 | *Gloeocapsa stegophila* | *Gloeocapsa* sp.4 |
| *Moss* | *Moss* | *Gloeocapsa magma* | *Gloeocapsa* sp.5 |
| *Microcoleus* sp. | *Microcoleus* sp. | | |

6. 3.1 change "population distribution" to "community distribution" (population represents genotypes of the same species!), and throughout the whole manuscript.

**Response:** Thank you pointing this out. "population distribution" has been changed to "community distribution", see **Line 285**. The entire document has been uniformly revised.

7. Line 190/191 – do not discuss your data in the results

**Response:** Thanks for your suggestion. The data discussions in the Results section have been moved to the Discussion section. The content from line190/191 has been moved to **Lines 671-672**.

8. 3.2 change "Biological Population Distribution" to "community distribution" and throughout the whole manuscript

**Response:** Thank you pointing this out. "population distribution" has been changed to "community distribution", see **Line 297**. The entire document has been uniformly revised.

9. Line 213 – do not discuss your data in the results

**Response:** Thanks for your suggestion. The data discussions in the Results section have been moved

to the Discussion section. The content from line213 has been moved to **Lines 671-672**.

10. Fig. 5a/b – delete Chinese letters

**Response:** Thank you for your suggestion! The Chinese letters in Figures 5a/b have been removed. See **Line 315**.

11. Fig. 7 e/f, 9f are of bad quality, and taxa cannot be identified!

**Response:** Thank you for this suggestion. The original images in Figures 7e/f and Figure 9f were of poor quality and have been re-taken or replaced with higher quality images to ensure clear identification of the taxonomic units. See the revised Fig. 9e/f on **Line 329** and Fig. 11f on **Line 347**.

12. Table 1 and 2 – although environmental data are very important for such a study, there are real data missing. Less or more sunlight does not tell the reader anything - real data instead would help! What was the temperature in the sun or at shaded sites?

**Response:** Thank you for your suggestion! The original Tables 1 and 2 lacked actual environmental data. Actual temperature data has been added; see Fig. 16d on **Line 422**, which shows the surface temperatures at different orientations of the Altar of Prayer for Good Harvests in the Temple of Heaven on a clear afternoon in April. The micro-environmental humidity test data was unstable (highly affected by wind) and the test failed. Further work is needed in the future to obtain this data.

13. In the discussion I miss more European studies. See also the recent papers of Patrick Jung on bioweathering mechanisms of rocks in the Atacama Desert by cyanobacteria and lichens!

**Response:** Thank you for introducing this excellent paper by Patrick Jung on the bioweathering mechanisms of cyanobacteria and lichens in the Atacama Desert. The paper has been cited in the introduction section on **Line 120** of the revised manuscript and discussed in detail in the discussion section from **Lines 683 to 701**.

14. I think the whole manuscript could be condensed (e.g. historical aspects in the introduction, number of figures etc.).

**Response:** Thank you for pointing this out. The historical background in the introduction has been streamlined; see the above section 2. The number of tables has been reduced from 5 to 3. However, the number of figures has increased from 20 to 24. The specific reasons are as follows:

(1) An overview photo of the study area is best included to provide a general observation and judgment of the distribution of biofilms on the rock surface. Therefore, Figure 1 has been added on **Line 86**.

(2) The data in this paper are primarily presented through figures, such as Fig. 2, 3, 5, 8, 10, 12, 15, 17, 18, and 20.

(3) The definitions of "Relative Volume" and "Relative Volume percentage" are more clearly and easily understood when illustrated, so Figure 4 has been added on **Line 254.**

(4) Since the species do not have specific names, but the discussion involves differences between species (distinguished by numbers, such as *Scytonema* sp.1 and *Scytonema* sp.2), the characteristics of different species need to be observed and distinguished through photos. Additionally, the morphology and color of the biological communities also need to be shown

in figures, such as Fig. 6, 7, 9, 11, 13, 14, and 21.

(5) A statistical chart showing the weathering degrees of the different orientations of the Altar of Prayer for Good Harvests is provided in Fig. 16 on **Line 422**. This helps to understand the differences in weathering at different orientations and compare the results of differential weathering indicated by the biological community analysis.

(6) Descriptions of various dissolution forms on the rock surface require corresponding photos for understanding, such as Fig. 22 on **Line 541** and Fig. 23 on **Line 556**.

(7) Fig. 19 on **Line 481** is more helpful in understanding the main population composition of different colored biological communities on the marble surface.

(8) Fig. 24 compares the growth of black biofilms on marble in Beijing and Florence. If the number of figures is considered too many, this figure can be removed, and the discussion text can be retained.

The manuscript does indeed contain a large number of figures. If reducing the number of figures would help with acceptance, I will consider how to streamline them. I apologize for this inconvenience.

**The author has made the following improvements:**

In addition to the modifications suggested by the reviewer, the author has made the following improvements:

1. The title has been changed from "Study on the Biological Communities and Bioweathering of Marble Surfaces at Temple of Heaven Park, Beijing, China" to "Study on the Biological Communities and Bioweathering of Marble Surfaces at the Altar of Prayer for Good Harvest in the Temple of Heaven (Beijing, China)." This change is more appropriate because the study only focuses on the bioweathering of marble surfaces at the Altar of Prayer for Good Harvest and does not cover other areas of the Temple of Heaven Park.

2. An overview photo of the study area has been added to provide a general observation and judgment of the distribution of biofilms on the rock surface (Fig. 1, **Line 87**). A more detailed discussion of the distribution patterns of biofilms on the marble surfaces in the study area has been included。 **Lines 28-38**

3. An introduction to the cloud Chi Heads at the Altar of Prayer for Good Harvest has been added (**Lines 193-208**), along with a statistical chart showing the weathering degrees of the different orientations of the cloud Chi Heads (Fig. 16, **Lines 399-422**). This helps to understand the differences in weathering at different orientations and compare the results of differential weathering indicated by the biological community analysis.

4. Additional instruments have been used in the fieldwork to measure the temperature and humidity of the rock surfaces (**Lines 224-230**).

5. The discussion section has been expanded to include the following content:
   (1) A commentary on the advantages and disadvantages of traditional morphological

identification and molecular biology methods for the special samples of biofilms on stone cultural relics (**Lines 561-581**).

(2) A commentary on the scientific validity of the visual analysis method for estimating the relative volume and relative percentage of species based on microscopic observations. (**Lines 582-589**).

(3) A more detailed discussion of the distribution regularities of biofilms on the marble surfaces in the study area (**Lines 590-609**).

---

## Author Response (AR2)

Dear Dr. Middelburg,

Thank you for your email and for accepting my paper for publication in Biogeosciences. I am pleased to hear that the manuscript is now accepted pending technical corrections. Below are the responses to the technical corrections you have requested.

I have made the necessary changes and hope that these revisions meet your requirements. Please let me know if there are any further adjustments needed.

Thank you for your time and consideration.

Best regards,
Dr. Tian
Professor of China University of Geosciences (Beijing)
E-mail: typ@cugb.edu.cn

Technical Corrections and Responses:

1. Figure 16 citation: Figure 16 is cited after figures 1 and 2, before figures 3 and further. It is common practice to number figures based on their appearance. Please renumber and correct all figure numbers throughout the document.
**Response**: The figure numbers have been modified as requested. However, based on the content of the text, it would be more appropriate to place Figure 16 after Figure 3. Since Figure 2 and Figure 3 both introduce the climate of the study area, their numbering should be consecutive.

2. Y-axis digits in Figures 2 and 3: Figures 2 and 3 have too many digits on the y-axis. For example, change 300.00 to 300, 40.00 to 40, etc.
**Response**: The y-axis digits have been modified as requested.

3. Line 271: (3) Relative Volume percentage….
**Response**: Not modified, as this follows the numbering from above (1). See **Line 261** in the revised manuscript 3.

4. Line 383: Replace "on the other hand" with "however" (in English, "on the one hand" and "on the other hand" always come together).
**Response**: The phrase has been modified as requested.

5. Line 489: … (Fig. 20). A conceptual diagram of the formation of these…
**Response**: The line has been modified as requested.

6. Line 576: … framework. This makes it difficult to ….
**Response**: The line has been modified as requested.

7. Line 609: … dust, which also needs to be taken into account.
**Response**: The line has been modified as requested.

8. Line 645: … facing. Consistently, the metabolic…
**Response**: I suggest modifying it to "… facing, while the metabolic…" as it indicates a contrast. The text means that the strongest weathering intensity of the Temple of Heaven is in the south direction, while the strongest weathering of the Florence Cathedral is in the southeast direction.